# p1/s1, a 3'-nucleotidase/nuclease, allows *Leishmania major* to circumvent host innate immune response mechanisms

Stella M. Schmelzle[1], Michaela Bergmann[1], Bianca Walber[1], Jamal Shamsara[2,3], Tanja Ziesmann[4,5], Ute Distler[4,5], Csaba Miskey[6], Liam Childs[7], Peter Kolb[2,3], Stefan Tenzer[4,5], Katrin Bagola[1], Ger van Zandbergen[1,4,5]*

**1** Division of Immunology, Paul-Ehrlich-Institut, Langen, Germany, **2** Institute for Pharmaceutical Chemistry, Philipps-Universität Marburg, Marburg, Germany, **3** Core Facility "Chemoinformatics & Molecular AI", Philipps-Universität Marburg, Marburg, Germany, **4** Institute for Immunology, University Medical Center, Johannes Gutenberg-University Mainz, Mainz, Germany, **5** Research Center for Immunotherapy (FZI), University Medical Center, Johannes Gutenberg-University Mainz, Mainz, Germany, **6** Division Haematology, Cell and Gene Therapy, Genomics Core Facility, Paul-Ehrlich-Institut, Langen, Germany, **7** Research Group Host-Pathogen Interactions, Paul-Ehrlich-Institut, Langen, Germany

\* ger.vanzandbergen@pei.de

## Abstract

3'-nucleotidases/nucleases, distinct class I nucleases of protozoan parasites, play a pivotal role in extracellular purine salvage. As *Leishmania* are purine auxotrophs and lack *de novo* synthesis, ectoenzymes facilitating nucleotide and nucleic acid cleavage are indispensable for subsequent uptake. Employing quantitative proteomics, we characterized a class I nuclease *p1/s1* cluster in *L. major* that comprises enzymes exhibiting dual 3'-nucleotidase and endonuclease activity. Expression of these enzymes is induced upon miltefosine or staurosporine treatment and was specifically detected in stationary-phase, but not in logarithmic-phase promastigotes. After confirming secretion of p1/s1, ecto-enzymatic activity was detected on parasites and in the culture supernatant. Viable null mutants deficient for the *p1/s1* cluster were only obtained when a diCre-based inducible knockout system was applied, whereas direct deletion approaches were lethal. The viable knockout strains exhibited significantly reduced 3'-nucleotidase/nuclease activity. Notably, these parasites adapted by compensatory enrichment of various alternative purine salvage proteins at the proteomic level. Furthermore, both enzymatic functions implied mechanisms of host-pathogen interactions to facilitate infection establishment: Utilizing 3'-nucleotidase activity, *Leishmania* generate extracellular adenosine to suppress inflammatory cytokine secretion from macrophages and reduce lymphocyte proliferation in a human primary cell model. The presence of ecto-nucleases also allowed these parasites to degrade and survive neutrophil extracellular traps, a potent first-line innate immune mechanism in pathogen defense. In summary, our integrative approach combining proteomics, immunological and genome editing methods expands current knowledge about *Leishmania major* 3'-nucleotidases/nucleases. By offering new insights into

**Data availability statement:** The mass spectrometry proteomics data have been deposited to the ProteomeXchange Consortium (http://proteomecentral.proteomexchange.org) via the jPOST partner repository with the dataset identifiers PXD066928 (ProteomeXchange) and JPST003974 (jPOST) (https://repository.jpostdb.org/entry/JPST003974.0). Whole genome sequencing reads of a parental strain and p1/s1 iKOs (L. majorCas9/T7/diCre/3'flox, L. majorΔp1/s1 clones E10 & F4) are available as SRA data at NCBI BioProject: PRJNA1330523 (https://www.ncbi.nlm.nih.gov/bioproject/PRJNA1330523).

**Funding:** This work was supported by LOEWE-Zentrum DRUID (D3 and B3 to GvZ; B3 and CF-Chemoinformatics & Molecular AI to PK) and the Deutsche Forschungsgemeinschaft (DFG) (SPP 2225 # 446496937 and 446605368 to GvZ; SPP 2225 #446605368 and SFB1292/2 #318346496 to UD). The funders had no role in study design, data collection and analysis, decision to publish, or preparation of the manuscript. No authors received a salary from any of the funders.

**Competing interests:** The authors have declared that no competing interests exist.

the diverse involvements in host-pathogen interactions, we highlight p1/s1 as pivotal factor during infection and potential drug target.

## Author summary

*Leishmania* parasites are purine auxotroph and depend on extracellular nucleotide and nucleic acid cleavage for subsequent uptake. In this study, we describe p1/s1, a secreted 3'nucleotidase-nuclease in *Leishmania major*, to be expressed in infectious stationary-phase but not logarithmic-phase promastigotes. Employing a quantitative proteomics dataset, we show that it is also induced upon drug-induced stress. Genomic deletion of the p1/s1 cluster by an inducible knockout approach reduced both 3'-nucleotidase and endonuclease activity and impaired the parasite's ability to survive NET-induced killing. However, proteomics data reveal that *L. major* partially compensates for the loss of p1/s1 by enriching other purine salvage-linked proteins, and seems to prevent drug-induced cell death. In a human primary macrophage infection model, p1/s1 nucleotidase activity metabolizes 3'-AMP to adenosine and phosphate which replicates the anti-inflammatory and lymphocyte proliferation suppression effects observed with adenosine. Collectively, these findings emphasize the relevance of 3'-nucleotidase/nuclease activity in driving *L. major* pathogenicity through its multiple molecular mechanisms.

## Introduction

Infection of mammals with the single-cell protozoan *Leishmania* causes the neglected tropical disease leishmaniasis. *Leishmania (L.) major*, an old-world species provoking cutaneous leishmaniasis, are exposed to complex environmental changes during its diphasic life cycle. They alternate between the motile promastigote stage in the sand fly digestive tract and the intracellular amastigote stage, residing in phagolysosomes of mammalian phagocytes, mainly macrophages. Like all trypanosomatids, *Leishmania* lack purine *de novo* biosynthesis pathways, are thus auxotrophs and dependent on salvage from their host [1, 2].

In order to mobilize extracellular nucleotides and nucleic acids for metabolism, trypanosomatids utilize distinct ecto-enzymes with dual 3'-nucleotidase/nuclease activity, followed by nucleoside transport across the membrane. These class I nucleases have been biochemically characterized in a wide range of species and are exclusive for protozoans, plants, and fungi [3]. They rely on five conserved domains for $Zn^{2+}$-mediated nucleophilic hydrolysis to cleave nucleic acids and 3'-nucleotides, with a preference for single-stranded substrates and 3'-AMP, respectively [3–6].

3'-nucleotidases/nucleases are often expressed in specific life stages and can be induced by nutrient starvation in different *Leishmania* species [7–10]. For *L. major*, the amastigote-specific LmaC1N [11] and a promastigote-specific 3'-nucleotidase/

nuclease [12] with membrane localization are described. p1/s1 (*LmjF.30.1500* and *LmjF.30.1510*), termed by its similarity to nucleases P1 of *Penicillium citrinum* and S1 of *Aspergillus oryzae* [5,13], is arranged in a genomic array with four gene copies of another putative class I nuclease (*LmjF.30.1460 – 90*). The latter one is differentially transcribed in *L. major* life stages, predominantly in amastigotes [14]. Specifically, p1/s1 has been detected in the secretome of *L. major* [15].

Such nucleotidase and nuclease ecto-activities suggest a dual role in both parasite metabolism and host-parasite inter-actions. 3'-AMP is an intermediate of the extracellular 2',3'-cyclic AMP pathway and is found in various tissues [16,17]. Hence, parasites can generate extracellular adenosine by cleaving 3'-AMP through their 3'-nucleotidase activity, which triggers anti-inflammatory responses in macrophages via $A_{2a}$ and $A_{2b}$ receptors, thereby promoting infection establishment [9,18]. Additionally, an endonuclease activity enables parasites to degrade extracellular DNA, facilitating escape from neutrophil extracellular traps (NETs) [19,20]. As neutrophils act as a "Trojan horse" in early infection [21], this evasion is crucial for *Leishmania* to avoid NET-mediated immune responses.

Given the obligatory nature of purine salvage in trypanosomatids, proteins of this pathway have been declared as promising candidates for drug development. Allopurinol, a nucleobase analogue, serves as an example already used in veterinary medicine for treating canine leishmaniasis [22]. The exploitation of purine nucleoside derivatives as antiproto-zoan agents supports this approach [23–25]. While the inhibition of 3'-nucleotidases/nucleases as a therapeutic strategy remains ambiguous due to the lack of specific inhibitors, the diverse roles of these enzymes at both the parasite and host levels demand further investigation.

Here, we investigated in detail the role of the ecto-3'-nucleotidase/nuclease p1/s1 in *L. major* promastigotes by employ-ing quantitative proteomics, genome editing and *in vitro* human infection models. Besides characterizing localization and life-stage specificity of its enzymatic activity, we observed p1/s1-dependent functions to obviate innate host immune responses. Additionally, we show how *L. major* compensated an induced *p1/s1* knockout by enhancing purine salvage protein expression.

## Methods

### Ethics statement

For working with human PBMCs, we obtained buffy coats of anonymized blood donors from the blood donation center (DRK-Blutspendedienst Hessen GmbH), Frankfurt am Main, Germany, which did not require approval by an Ethics committee.

### *L. major* parasite culture

*Leishmania major* (MHOM/IL/81/FEBNI) promastigotes were cultured on a semi-solid rabbit blood-agar system as described before [26]. Cultures were seeded on day 0 with 50,000 viable *L. major* in 100 µl per well of a 96-well plate. Logarithmic-phase (logPh) promastigotes, 3 days post passaging, or stationary-phase (statPh) promastigotes from 6-8 days post passaging were used for experiments. Stationary-phase promastigotes typically consist of approximately 2–10% metacyclic parasites and 50–60% dead parasites exhibiting apoptosis-like features, e.g., phosphatidylserine expo-sure (Fig A in S1 Appendix).

Axenic amastigotes were generated as described in [27] and washed in tissue culture plates prior to experiments. Para-sites either expressed DsRed as fluorescent marker or a Cas9/T7 RNA polymerase [28]/ dimerizable Cre (diCre) construct [29]. Parasites were cultured with respective selection markers: 30 µg/ml Hygromycin B Gold, 10 µg/ml Blasticidin, 40 µg/ml G418, 40 µg/ml Puromycin (all InvivoGen), 100 µg/ml Bleomycin (Jena Bioscience).

### *L. major* proteomics

**Sample preparation.** $25 \times 10^6$ viable *L. major* promastigotes were lysed in 40 µL SDS-containing buffer (2% (w/v) SDS, 5 mM DTT, 1x cOmplete Protease Inhibitor Cocktail-EDTA (PIC, Roche), 50 mM HEPES pH 8.0). Seven biological

replicates from consecutive passages were taken. To promote lysis, samples were incubated at 95°C for 5 min followed by sonication for 15 min (30 s on/off cycles, high power) at 4 °C using a Bioruptor device (Diagenode, Liège, Belgium). The protein concentration was determined using the Pierce 660 nm protein assay (Thermo Fisher Scientific) according to the manufacturer´s protocol. Subsequently, proteins (corresponding to 10 µg) were digested using single-pot solid-phase-enhanced sample preparation (SP3) as described in detail before [30], [31]. In brief, proteins were reduced and alkylated using DTT and iodoacetamide (IAA), respectively. Afterwards, 2 µL of carboxylate-modified paramagnetic beads (Sera-Mag SpeedBeads, GE Healthcare, 0.5 µg solids/µL in water as described by [30] were added to the samples. After adding acetonitrile to a final concentration of 70% (v/v), samples were allowed to settle at room temperature for 20 min. Subsequently, beads were washed twice with 70% (v/v) ethanol in water and once with acetonitrile. Beads were resuspended in 50 mM $NH_4HCO_3$ supplemented with trypsin (Trypsin Gold, Mass Spectrometry Grade, Promega) at an enzyme-to-protein ratio of 1:25 (w/w) and incubated overnight at 37 °C. After overnight digestion, acetonitrile was added to the samples to reach a final concentration of 95% (v/v) followed incubation at room temperature for 20 min. To increase the yield, supernatants derived from this initial peptide-binding step were additionally subjected to the SP3 peptide purification procedure as described before [30]. Each sample was washed with acetonitrile. To recover bound peptides, paramagnetic beads from the original sample and corresponding supernatants were pooled in 2% (v/v) dimethyl sulfoxide (DMSO) in water and sonicated for 1 min. After 2 min of centrifugation at 12,500 rpm and 4 °C, supernatants containing tryptic peptides were transferred into a new vial for MS analysis and acidified with 0.1% (v/v) formic acid.

Alternatively, pellets were resuspended in 7 M Urea, 2 M Thiourea, 2% CHAPS subjected to tryptic digestion and processed as in [32].

**Liquid-chromatography mass spectrometry (LC-MS).** Peptides were analyzed by LC-MS using an Evosep One chromatography system (Evosep, Odense, Denmark) coupled online to a timsTOF SCP mass spectrometer (Bruker Corporation, Billerica, MA, USA). For MS-analysis, samples were loaded onto EvoTips according to the manufacturer´s protocol. Subsequently, peptides were separated using a reversed phase C18 column (Aurora ELITE UHPLC emitter column, 15 cm x 75 µm 1.7 µm, IonOpticks) and the "Whisper Zoom 40 samples per day (40SPD)" method provided by the manufacturer. Column was heated to 50°C. Mobile phase A was 0.1% FA (v/v) in water and mobile phase B 0.1% FA (v/v) in ACN. Eluting peptides were analyzed in positive mode ESI-MS using parallel accumulation serial fragmentation (PASEF) enhanced data-independent acquisition mode (DIA) [33]. The dual TIMS (trapped ion mobility spectrometer) was operated at a fixed duty cycle close to 100% using equal accumulation and ramp times of 100 ms each spanning a mobility range from $1/K_0 = 0.7$ Vs cm$^{-2}$ to 1.3 Vs cm$^{-2}$. We defined 14 isolation windows with variable width covering a precursor mass range from $m/z$ 307–1,198. Window scheme was generated using the python package py_diAID [34] resulting in seven diaPASEF scans per acquisition cycle (total cycle time of 0.85 s). The collision energy was ramped linearly as a function of the mobility from 59 eV at $1/K_0 = 1.6$ Vs cm$^{-2}$ to 20 eV at $1/K_0 = 0.6$ Vs cm$^{-2}$.

Alternatively, samples were measured by high-resolution nanoUPLC separation and analyzed by data-independent acquisition on the Waters Synapt G2-S platform [32].

All samples were measured in triplicates.

**Raw data processing.** MS raw data were processed using DIA-NN (version 1.9.2) [35] applying the default parameters for library-free database search. Data were searched using a custom compiled database containing UniProtKB/TrEMBL entries of the *Leishmania major* proteome and a list of common contaminants (version release January 2025, 8,537 entries, Taxon ID 5664). For peptide identification and in-silico library generation, trypsin was set as protease allowing one missed cleavage. Carbamidomethylation was set as fixed modification and the maximum number of variable modifications was set to zero. The peptide length ranged between 7–30 amino acids. The precursor $m/z$ range was set to 300–1,800, and the product ion $m/z$ range to 200–1,800. As quantification strategy we applied the "QuantUMS (high accuracy)" mode with RT-dependent median-based cross-run normalization enabled. We used the build-in algorithm

of DIA-NN to automatically optimize MS2 and MS1 mass accuracies and scan window size. Peptide precursor FDRs were controlled below 1%.

Alternatively, processing of raw data was accomplished by the ProteinLynx Global Server for *Leishmania* reference proteome and quantified using the software pipeline ISOQuant [32].

**Statistical and downstream data analysis.** In the final proteome datasets [36], proteins had to be identified by at least two peptides. Statistical analysis of the data was conducted using Student's t-test, which was corrected by the Benjamini-Hochberg (BH) method for multiple hypothesis testing (FDR of 0.01). Additionally, differentially expressed proteins had to display a log2(FC) of > 1.5/< -1.5, a p-value < 0.05 and needed to be present in at least four samples in one condition of the comparison used for analysis of cell-death induced L. major, or a log2(FC) of > 0.5/< -0.5, a p-value < 0.01 and needed to be present in at least 16 samples for the analysis of the p1/s1 mutants.

In order to plot proteins that were exclusively detected in only one condition, we set an arbitrary value.

## Nucleotidase assay

Nucleotidase activity of $2 \times 10^6$ intact *L. major* was determined by the release of free phosphate ($P_i$) from 3'-AMP or 5'-AMP (both SigmaAldrich). After 3 hours of parasite incubation in 100 µl reaction buffer (116 mM NaCl, 5.4 mM KCl, 5.5 mM D-Glucose, 50 mM HEPES pH 7-7.6) [37] at 27 °C, parasites were separated into pellet and supernatant by centrifugation and 100 µM substrate added for 1 hour at 27 °C. Intact pelleted parasites represented membrane-bound ecto-nucleotidase activity, the supernatant fraction secreted ecto-nucleotidase activity. Released $P_i$ was quantified using a malachite green based phosphate assay kit (SigmaAldrich) according to instructions. $A_{620nm}$ was measured with a ClarioStarPlus plate reader (BMG Labtech). A sample containing only substrate was used as blank, a control containing only parasites was always checked for $P_i$ carryover. All reagents were tested for unspecific $P_i$ content before use. Nucleotidase activity was expressed as µM released $P_i / 2 \times 10^6$ *L. major*/h.

## Endonuclease assay

Nuclease activity of $2 \times 10^6$ intact *L. major* was determined by visualizing enzymatic degradation of a DNA substrate on a 0.8% TAE-agarose (BioRed) gel stained with GelRed (biotium). After 3 hours of parasite incubation in 100 µl reaction buffer (30 mM sodium acetate pH 5.3, 100 mM NaCl, 2 mM $ZnCl_2$) [7] at 27 °C, parasites were separated into pellet and supernatant by centrifugation and 500 µg substrate added for 1 hour at 27 °C. Intact pelleted parasites represented membrane-bound ecto-nuclease activity, the supernatant fraction secreted ecto-nuclease activity. For endonuclease activity, single-stranded or double-stranded circular M13 phage genome (New England Biolabs) was used as substrate. For nuclease activity on NETs, respective NET-enriched supernatant was used. Reaction times were set as specified. Remaining DNA substrate was semi-quantified on gel images relative to a 500 µg undigested control using ImageJ 1.53c. A sample containing substrate and DNaseI (Invitrogen) was used for positive digestion control on every gel. Relative nuclease activity was calculated as difference to the undigested substrate.

## His$_6$ enrichment and Western blot

$12 \times 10^6$ *L. major* logPh promastigotes were lysed in 20 µl Laemmli buffer (500 mM Tris/HCl pH 6,8, 38% glycerin, 10% SDS, 0.6 M DTT, 0,01% bromophenol blue) [38]. Corresponding culture supernatant was incubated with 15 µl HIS-Select nickel magnetic agarose beads (SigmaAldrich) for 30–60 min at room temperature in an overhead shaker and washed twice in PBS. Proteins were eluted in 20 µl Laemmli buffer at 65 °C for 10 min.

Protein samples were denatured for 5 min at 95 °C and separated on a 12% SDS-PAGE. After blotting, PVDF membranes were incubated with 1 µg/ml anti-αTubulin (GeneTex GTX628802) and 776 ng/ml anti-His$_6$ (Cell Signaling Technology 2366) antibodies overnight at 4 °C, followed by 1 hour at room temperature with HRP-conjugated anti-mouse

secondary antibody (Santa Cruz sc516102). The membrane was incubated with Immobilon Forte Western HRP substrate (Millipore) for 1 min at room temperature, signal was visualized in an ECL Chemostar (Intas).

## Human monocyte-derived macrophage (hMDM) generation

Monocytes were isolated from peripheral mononuclear cells (PBMCs) of anonymized healthy human donors as described before [39]. Buffy coats were provided by the DRK-Blutspendedienst Hessen GmbH (Frankfurt am Main). CD14$^+$ monocytes were isolated by magnetic cell sorting (Miltenyi Biotec) and differentiated with 50 ng/ml M-CSF (R&D Systems) or 30 ng/ml GM-CSF (Sanofi) for 5–7 days at 37 °C, 5% $CO_2$ in a humidified atmosphere. Further activation was done with 50 ng/ml IFNγ (Sigma) for 24 hours.

## Autologous peripheral blood lymphocyte (PBL) isolation

CD14$^-$ PBMCs were stored at -80 °C for the time of macrophage differentiation and thawed upon usage [39].

## Isolation of polymorphonuclear cells (PMNs)

PMNs of healthy individuals were isolated from fresh blood as described in [40] or from buffy coats as previously described [26]. 12x10$^6$ cells were seeded on poly-D-lysine (Gibco) coated 12-well plates. Neutrophil extracellular trap (NET)-enriched supernatants were induced and harvested as reported before [41]. DNA content was determined using a Quant-it RiboGreen assay (Invitrogen).

## Determination of hMDM infection

Differentiated macrophages were infected with *L. major*$^{DsRed}$ promastigotes from the statPh at a multiplicity of infection (MOI) of 10 for 3 hours at 37 °C before remaining extracellular parasites were washed off. After the specified time post infection (pi), the infection rate of viable single cells was quantified by detecting the parasite's fluorescence marker by flow cytometry on either LSR Fortessa or Symphony A3 Cell Analyzer (both BD Bioscience). Parasite burden was determined as DsRed mean fluorescence intensity (MFI) of single, viable, infected macrophages. Cell viability was monitored using staining with 1:1000 ZombieAqua (BioLegend).

Alternatively, cells were fixed in 4% PFA (Sigma) and permeabilized using 0.5% saponin (AppliChem PanReact), then intracellular parasites were stained with 1:10,000 anti-*L. major* rabbit serum and 1:1000 goat anti-rabbit AlexaFluor 568 secondary antibody (Invitrogen A-11036).

If applicable, adenosine (SigmaAldrich) or 3'-AMP (Santa Cruz Biotechnology) were added during infection at indicated concentrations.

## Analysis of PBL proliferation in a co-culture with hMDM

Macrophages and PBLs were stained in 1 μM CellTrace Far Red (Invitrogen) for 20 min at 37 °C. 0.2 × 10$^6$ stained macrophages were infected with *L. major* promastigotes from the statPh at MOI = 20 for 3 hours at 37 °C before 1x10$^6$ stained PBLs were added. Infection was carried out in U-shaped 96-well plates with 2x10$^4$ hMDMs per well and 10 technical replicates per condition which were pooled before analysis. PBL proliferation was analyzed by flow cytometry 4 days pi as CellTrace$_{low}$ fraction of viable, single PBLs. Infection rate and parasite burden of hMDM were determined as stated before. Cell viability was monitored using staining with 1:1000 ZombieAqua (BioLegend).

If applicable, adenosine or 3'-AMP were added together during infection at indicated concentrations.

## Neutrophil killing assay

Isolated granulocytes were tested for activation status by CD66b and CD62L staining beforehand. 2x10$^6$ PMNs in a 24-well plate were treated with 10 μg/ml Cytochalasin D (SigmaAldrich) for 20 min at 37 °C. 130 U DNaseI were added

at the same time, if applicable. *L. major* statPh promastigotes were added at MOI = 1 and incubated for 2 hours at 37 °C and then transferred to 27 °C for two days. Corresponding controls containing reagents and parasites but no PMN were treated the same. Motile, viable parasites were counted in a Neubauer chamber and numbers normalized to the respective control gave the percentage of viable parasites.

## ELISA

Cell free supernatants from infected samples were analyzed for soluble cytokine content using enzyme-linked immunosorbent assay (ELISA) (DuoSet, R&D Systems) following instructions. TNFα content was measured as 1:12.5 dilutions. Signals were developed using horseradish peroxidase substrate solution (BioLegend) and stopped by addition of 2 N $H_2SO_4$ (Carl Roth). Absorbance was measured at 450 nm using 570 nm as reference wavelength at a ClarioStarPlus. Total cytokine concentration was determined by 4-parameter regression of the standard row in Mars v4.00. Samples were measured in duplicates.

## Staining of surface markers

Human primary cells were stained for surface markers in PBS pH 7.2 plus 0.5 g/L BSA and incubated for 30 min at 4 °C. Antibodies were added accordingly: anti-$A_{2a}$-AlexaFluor488, anti-$A_{2b}$-AlexaFluor647 (both R&D Systems, 1 µg/1 × 10^6 cells). Cells were also stained with 1:1000 ZombieAqua (BioLegend) or 1 µM SYTOXBlue (Invitrogen) as viability marker. Expression was analyzed by flow cytometry as percentage positive cells.

## Generation of *L. major* inducible knockout strains

*L. major* expressing Cas9/T7 RNA polymerase and a dimerizable Cre (diCre) construct were used to integrate a 3' and a 5' flox cassette in two consecutive steps by CRISPR/Cas9-facilitated homologous recombination, adapting the method of [29]. Each cassette was amplified using 0.5 µM forward and reverse primer (sequences see Table 1), 15 ng/µl plasmid template (pGL2314 for 3' flox [29], pTNeo-eGFP-loxP (modified based on [28]) for 5' flox), 4 µl GC Enhancer, 10 µl Platinum Mastermix (both Invitrogen) and 3 µl $H_2O$ in a S1000 Thermal Cycler (BioRad) set to 35 cycles of 10 s at 98 °C and 3 or 4 min (3' flox/ 5' flox) at 72 °. Single guide RNA (sgRNA) templates were amplified using 2 µM primer, 2 µM scaffold, 5 µl PWO Mastermix (Roche) and 1 µl $H_2O$ for 35 cycles of 10 s at 98 °C, 30 s at 60 °C and 15 s at 72 °C. Products were checked on a 1% agarose gel and sterilized for 5 min at 94 °C.

60x10^6 promastigotes from the logPh were pulsed once with the PCR products in Human T-cell Nucleofector buffer (Lonza) using a Nucleofector 2b (Lonza) with program X-001. Electroporated cells were incubated overnight in 9 ml *L. major* medium at 27 °C. After addition of the respective antibiotics and 10% fetal calf serum (SigmaAldrich), parasites were distributed in 100 µl on a 96-well rabbit blood agar plate. After 14–21 days, grown clones were checked for mCherry (3' flox) and eGFP (5' flox) fluorescence using flow cytometry and single cell clones were generated by serial dilution.

Recombination in full-floxed strains was induced by addition of 100 nM rapamycin (SigmaAldrich) and fluorescence monitored over 9 days. Single cell clones were generated as above and checked for *p1/s1* and *eGFP* status by PCR (2 µl genomic DNA, 1.5 µl forward and reverse 10 µM primer each, 5 µl PWO Mastermix) by 3 min at 95 °C and 25 cycles of 30 s at 95 °C, 30 s at 59 °C, 45 s at 72 °C plus final extension for 10 min at 72 °C. Amplified products were checked on a 1% agarose gel.

## Generation of *L. major* addback strains

*p1/s1* (*LmjF.30.1500*) was inserted into the multiple cloning site of pLexLm plasmid, a plasmid based on pLEXSY-ble2.1 (Jena Bioscience, Cat. No. EGE-271) in which the rRNA promotor region of *L. donovani* has been exchanged by an *L. major* rRNA promotor region. Identity of the insert as *LmjF.30.1500* was confirmed by Sanger sequencing.

**Table 1. Oligonucleotide sequences.**

| Oligonucleotide | 5'-3' sequence |
| --- | --- |
| 5' flox cassette forward | CATTAACGTTTGGGAGACCCCTGTGCAAAGG-TATAATGCAGACCTGCTGC |
| 5' flox cassette reverse | AGGCGCGGGCGCATGGGCACGTGGAAAAAGC-CAATTTGAGAGACCTGTGC |
| 5' sg template | GAAATTAATACGACTCACTATAGGAACCGTCGG-CGATAAAAAATGTTTTAGA GCTAGAAATAGC |
| 3' flox cassette forward | GCCGCTCGAGAGCCTTTATGTTTGTGGCTTGC-CCGATTACGCATGATCTA |
| 3' flox cassette reverse | GCCGAAAAGAACACAAGAAAAAAATCTTGCGAG-CAAAGAAAGCTGGGTTCCATGG |
| 3' sg template | GAAATTAATACGACTCACTATAGGATTGTCGT-GATGGGTTCTGGGTTTTAGAGCTAGAAATAGC |
| *p1/s1 (LmjF.30.1500/10)* forward | TCCGTCTGCCTCTCACAGTGCT |
| *p1/s1 (LmjF.30.1500/10)* reverse | CTCCTCGTGAATCGCCACCA |
| *eGFP* forward | CGTGACCACCCTGACCTACG |
| *eGFP* reverse | GGCGGATCTTGAAGTTCACCTTG |
| Scaffold [28] | AAAAGCACCGACTCGGTGCCACTTTTTCAAGTT-GATAACGGACTAGCCTTATTTTAACTTGC-TATTTCTAGCTCTAAAAC |

For addback strain generation, $60 \times 10^6$ *L. major*$^{\Delta p1/s1}$ clone E10 logPh promastigotes were electro-porated with SwaI (New England Biolabs) linearized and sterilized plasmid. Electroporated cells were incubated overnight in 9 ml *L. major* medium at 27 °C. After addition of bleomycin and 10% FCS, parasites were spread in 100 µl on a 96-well rabbit blood agar plate. After 14–21 days, grown clones were checked for *p1/s1* status using PCR and single cell clones were generated.

### Generation of *L. major* strains overexpressing p1/s1-His$_6$

*p1/s1* (*LmjF.30.1500*) was inserted into the multiple cloning site of pLexLm plasmid including an in-frame fusion with the encoded His$_6$-tag. Identity of the insert as *LmjF.30.1500* was confirmed by Sanger sequencing.

For generation of strain overexpressing *p1/s1*-His$_s$, $60 \times 10^6$ *L. major*$^{Cas9/T7/}$eGFP logPh promastigotes were electropo-rated with SwaI (New England Biolabs) linearized and sterilized plasmid. To generate strains overexpressing *p1/s1*, a *L. major*$^{Cas9/T7/eGFP}$ strain was used. Electroporated cells were selected and cultured as described above for *L. major* addback strains.

### Cell death assays

*L. major* logPh promastigotes were treated with increasing concentrations of miltefosine (Calbiochem) for 48 h, and para-sites were used in different assays to quantify different hallmarks of cell death:

Metabolic rate of $5 \times 10^6$ promastigotes was determined by colorimetric turnover of 1.1 mM Thiazolyl blue tetrazolium bromide (MTT, abcam) in phenolred-free RPMI (SigmaAldrich) for 4 hours at 27 °C. Formazan crystals were dissolved in DMSO (AppliChem PanReact) and absorbance was measured at 540 nm, using 700 nm as reference wavelength, at a ClarioStarPlus. Values were blanked to a sample containing parasites but no MTT. Relative cell metabolism representing viability was calculated by normalizing to the DMSO control.

$2 \times 10^6$ promastigotes were stained with 1:1000 AnnexinV-FITC (Miltenyi Biotec) in Ringer solution (B. Braun) for 20 min at 4 °C. Labelling was analyzed by flow cytometry, gating was set using a control stained in calcium-free PBS.

$2 \times 10^6$ promastigotes were permeabilized in 70% (v/v) EtOH for 30 min at -20 °C and stoichiometrically stained for DNA content using 5 µM SYTOXGreen (Invitrogen) in PBS containing 50 mM EDTA pH 7.4 and RNase (Thermo Fisher) for 30 min at 37 °C. Parasites were analyzed by flow cytometry on LSR II SORP (BD Bioscience), gating was determined on an unstained control. The share of cell cycle phases was determined based on DNA content.

Effective concentration (EC)$_{50}$ values were calculated based on 4-parameter regression in GraphPad Prism with constraints for bottom and EC$_{50} \geq 0$ and stated as ± 95% confidence interval (CI).

## Statistics

All graphs display mean ± standard deviation and were generated in GraphPad Prism v9.5.0. Significance was determined using a two-sided Welch's t-test with * $p \leq 0.05$, ** $p \leq 0.01$, *** $p \leq 0.001$, **** $p \leq 0.0001$.

## Results

### A set of 9 proteins, including p1/s1, shows increased abundance in *L. major* upon various cell death pressure

Identification of stress-related proteins in *Leishmania* is of great interest as they can feature potential drug targets [42–44] or be involved in mediating apoptotic-like cell death, a mechanism used by *Leishmania* to promote disease development and circumvent effective adaptive immune responses [39,45]. In order to designate such proteins in *L. major*, we applied stress to the parasites by treatment with low doses of miltefosine, an antileishmanial drug, or staurosporine, an unspecific kinase inhibitor, that would lead to cell death after 24 hours. Following treatment with miltefosine for 3 hours or staurosporine for 6 hours, with early cell death processes being induced (Fig A in S1 Appendix) but not being lethal at these time points, we performed a quantitative proteome analysis to compare protein expression. Volcano plots of proteins that show significantly increased abundance in the treated or untreated conditions are displayed in Fig 1A and 1B. Here, proteins with increased abundance in one condition that is displayed by a segregated log2 fold change, had not be detected in the other condition. Additionally, stationary-phase (statPh) promastigotes, harboring parasites that undergo intrinsic cell death, were analyzed and compared to untreated logPh parasites (Fig 1C). By choosing these conditions, proteins of immediate stress responses prior to cell death or survival promotion can be detected in the intersection of more abundant hits (Fig 1D). Proteins involved in drug-specific responses or metacyclogenesis of the statPh are solely detected in single conditions and are thus excluded.

Across the three comparisons, 4116 proteins were detected in total, corresponding to 51.73% of the reference *Leishmania* proteome. Taking the intersection of hits with increased abundance, only five proteins were commonly more abundant in all three comparisons, four were exclusively detected in parasites upon cell death induction or statPh promastigotes, but not in the viable control samples (Fig 1D and Table 2). Among these hits are several proteins involved in protein biogenesis and turnover but also proteins of diverse predicted functions. One of the exclusively detected hits, p1/s1, is the only protein with predicted enzymatic activity, namely a class I nuclease, and with proposed involvement in host-parasite interactions. In *Leishmania*, such enzymes are involved in essential purine salvage by eliciting 3'-nucleotidase and endonuclease activity and are presumably involved in cellular functions as well as in host-parasite interactions (reviewed in [48,49]). Given p1/s1 expression in stress-induced and infectious statPh *L. major*, this hit stands out as central link between parasite metabolism and disease progression. Studying its multiple functions and its potential role in parasite cell death could establish p1/s1 as potent drug target.

### Ecto-nucleotidase and -endonuclease activity of *L. major* increase from the logarithmic to stationary growth phase

Indeed, detected peptides assigned to p1/s1 can derive from six open reading frames arranged in a tandem array (*LmjF.30.1460* to *LmjF.30.1510)*. Sequence details were retrieved from EnsemblProtists release 61. All open reading

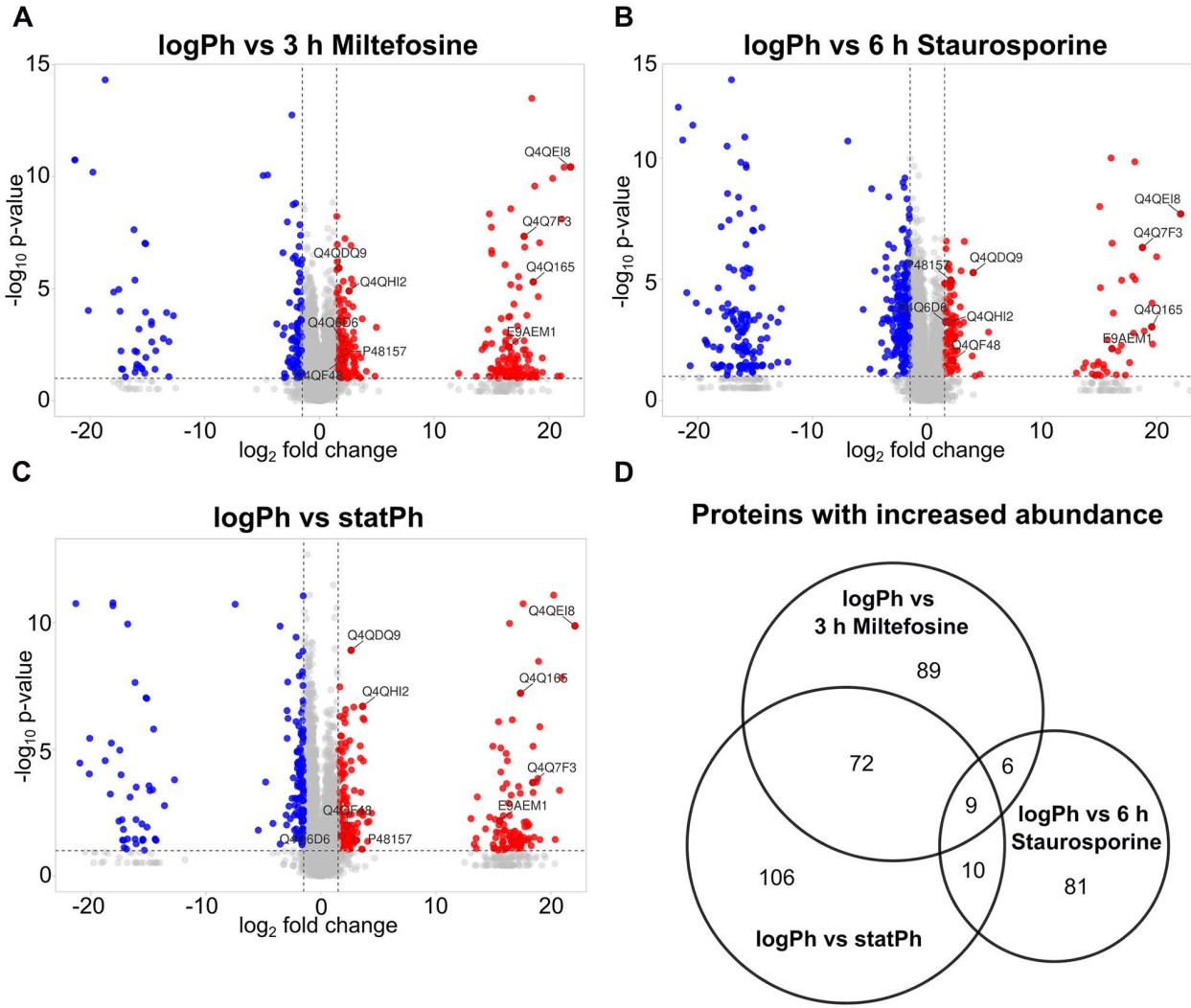

**Fig 1. Quantitative proteomic analysis of cell death-induced *L. major*. A**- Volcano plot of the quantified proteome of *L. major* treated with 25μM miltefosine for 3h, **B**- 25μM staurosporine for 6h and **C**- statPh promastigotes, always compared to untreated, logPh promastigotes. **A-C**- Volcano plot visualized with VolcaNoseR [46], proteins with commonly increased abundance are labelled (Table 2). Cutoffs were set at $\log_2 FC < -1.5, > 1.5$ and p-value < 0.01. Visible gaps in the fold change separate protein hits that revealed increased abundance in one condition but had not been detected in the other condition. In order to plot these proteins, we set an arbitrary value for each undetected protein. **D**- Venn diagram displaying the number of proteins with significantly increased abundance in each labeled condition versus the untreated logPh control. Intersections represent proteins detected as more abundant in several conditions. Template created using BioVenn [47]. Analysis based on 7 biological replicates and 3 technical replicates each.

frames share a common C-terminal domain. Whereas the four genes *1460–90* contain a putative transmembrane domain (TMD), *1500* and *1510* code for an N-terminal signal sequence for secretion (Fig 2A), as predicted by DeepLoc 2.1 [50]. We verified secretion by overexpressing a C-terminally His6-tagged *LmjF.30.1500* construct in *L. major*^Cas9/T7/eGFP. Parasite lysate was directly used for SDS-PAGE, whereas the corresponding culture supernatant was subjected to His6 enrichment over magnetic Ni-NTA beads. This eluate was also separated by SDS-PAGE. p1/s1-His6 and the housekeeping protein α-Tubulin were detected by Western blot. Whereas no His6-tagged proteins were detected in the lysate, a His6-tagged protein approximately at the predicted size of p1/s1 (35.1 kDa including signal sequence) was visible in the culture supernatant for both overexpressing clones H9 and F3 (Fig 2B), confirming p1/s1 secretion.

**Table 2. Common proteins with increased abundance in** *L. major* **undergoing cell death; annotations extracted from UniProt [49]. All peptides detected for p1/s1 could be assigned to the conserved C-terminal region of proteins in the cluster (Fig 2A).**

| UniProt code | UniProt annotation | Remark |
|---|---|---|
| Q4Q7F3, Q4Q7F4, Q4Q7F5, Q4Q7F6, Q4Q7F7, Q4Q7F8 | p1/s1 nuclease | Not detected in viable control, exclusive to cell-death related conditions |
| Q4QEI8 | Elongation factor 1α | |
| Q4Q165 | Putative polyubiquitin | |
| E9AEM1 | Arrestin-like N-terminal domain-containing protein | |
| P48157 | Large ribosomal subunit protein uL5 | Increased abundance in statPh *L. major* and upon cell death induction |
| Q4QDQ9 | Mitochondrial carrier protein | |
| Q4QF48 | EF-hand domain-containing protein | |
| Q4Q6D6 | p-glycoprotein e | |
| Q4QHI2 | Putative folate/biopterin transporter | |

To identify the putative homologous active site of p1/s1, AlphaFold-predicted structures of proteins coded in the *LmjF.30.1460 -1510* cluster were superimposed on homologous template structures, neglecting the putative signal sequence (Fig 2C). Based on homology to Endonuclease 2 from *Arabidopsis thaliana* (3W52)*,* five regions were identified to participate in the p1/s1 active site (Fig 2D), coinciding with previous reports on the *L. donovani* 3'-nucleotidase/nuclease homologue *Ld*Nuc$^S$ [8]. In contrast to the putative membrane-bound proteins Q4Q7F8-F5, encoded by *LmjF.30.1460-90*, only the secreted enzymes Q4Q7F4-3, encoded by *LmjF.30.1500-10*, contain this active site. Beyond p1/s1, further enzymes (Q66VY6, Q4QGQ3, Q4Q630) possess a homologous active site, indicating a putative functional redundancy.

Further, to assess overall 3'-nucleotidase activity arising from ecto-enzymes of *L. major,* and to compare p1/s1 expression and 3'-nucleotidase activity patterns, we quantified the release of free phosphate ($P_i$) from 3'-AMP. Intact parasites, harboring all membrane-bound ecto-enzymes, were separated from the supernatant, which contains secreted enzymes. Besides the secreted p1/s1, the *L. major* genome codes for additional 3'-nucleotidases/nucleases predicted to be membrane-localized. Comparing logPh and statPh promastigotes, a significant increase in 3'-nucleotidase activity in both the pellet and the supernatant fraction was observed, corresponding well to the elevated p1/s1 protein levels in statPh (Fig 1C). Overall, activity associated with pelleted parasites was higher (Fig 2E), likely related to non-secreted 3'-nucleotidases/nucleases. In contrast, degradation of 5'-AMP through 5'-nucleotidases was generally much lower than 3'-nucleotidase activity (Fig 2F).

Accordingly, ecto-endonuclease activity was determined using double-stranded M13 phage genome (dsM13) as substrate. Its degradation was visualized on an agarose gel for semi-quantification. Similar to 3'-nucleotidase activity, endonuclease activity is increased in statPh and a higher activity was observed on pelleted parasites than in the supernatant (Fig 2G and 2H). These results align with the pattern seen for 3'-nucleotidase activity and p1/s1 protein expression, suggesting both functions arise from p1/s1.

### Presence of 3'-AMP during *L. major* hMDM *in vitro* infection mimics anti-inflammatory effects of extracellular adenosine

As shown in Fig 2E, *L. major* statPh promastigotes can hydrolyze extracellular 3'-AMP to adenosine and $P_i$. Extracellular adenosine is a well-characterized anti-inflammatory stimulant of immune cells. GM-CSF-differentiated macrophages

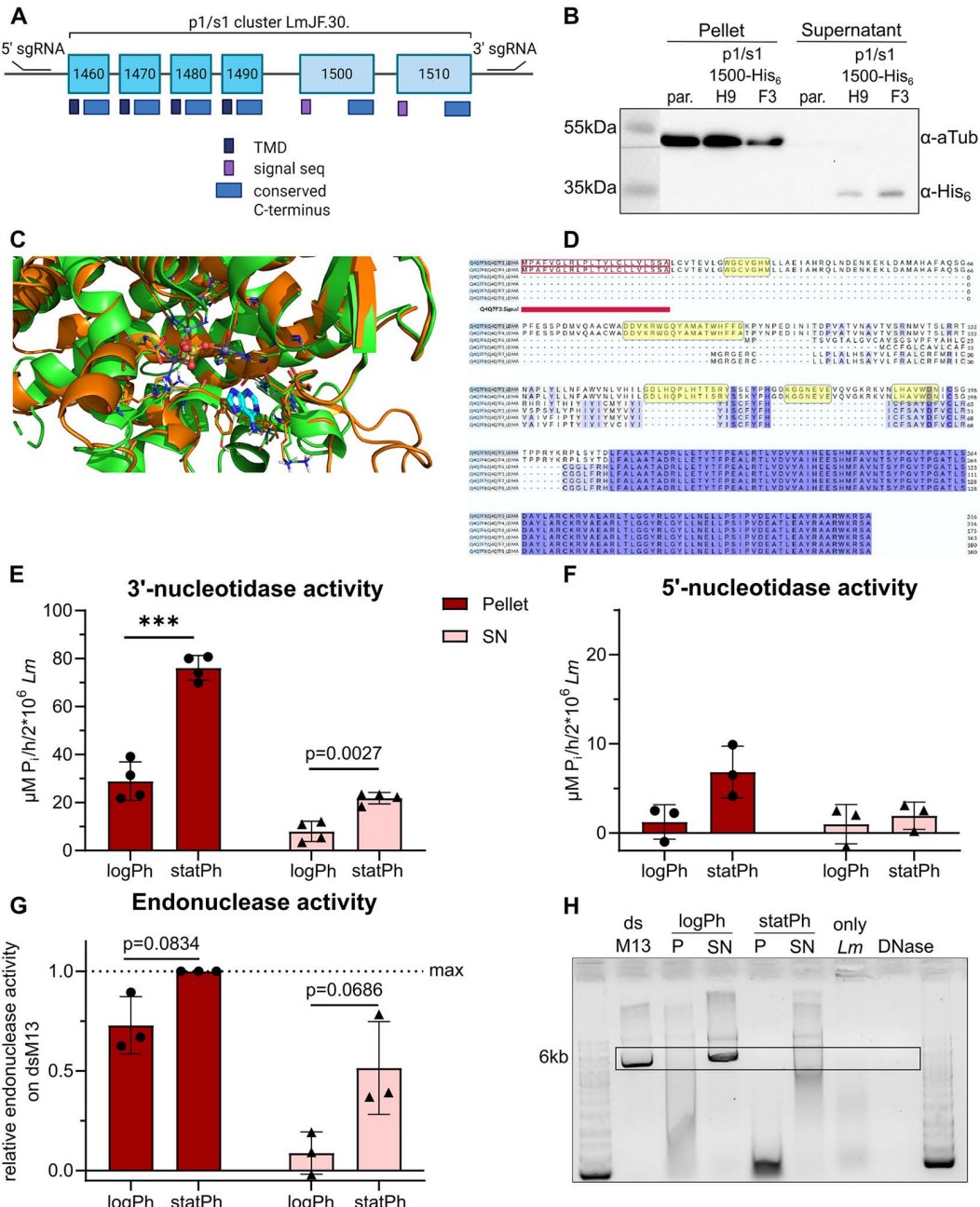

**Fig 2. Ecto-nucleotidase and -endonuclease activities of *L. major* increase in the stationary-phase. A**- Depiction of the genomic *p1/s1* cluster of six genes (*LmjF.30.1460 – 1510*) sharing a conserved domain but encoding either a signal sequence or a transmembrane domain (TMD). Indicated 5'sgRNA and 3'sgRNA were used for CRISPR Cas9-mediated inducible knockout of the cluster. Created in BioRender. Schmelzle, S. (2026) https://BioRender.com/mlghi1j. **B**- Western blot to determine p1/s1-His$_6$ localization in 12x10$^6$ *L. major* promastigotes overexpressing a C-terminally His$_6$-tagged *LmjF.30.1500* construct (clones H9 and F3) and a control parental strain (par.). Secreted, tagged proteins were enriched with magnetic Ni-NTA beads from the corresponding culture supernatant. a-Tubulin was used as loading control for pelleted parasites. Representative blot of n = 3. **C**- Superimposition of the AlphaFold-predicted structure of Q4Q7F3 (orange, encoded by *LmjF.30.1510*) onto the experimentally determined structure of a homologous enzyme in *Arabidopsis thaliana* (green; PDB ID: 3W52). The active site residues and the bound adenine base (cyan) in 3W52 are shown as sticks. Coordinated zinc ions are represented as gray spheres, and the sulfate ion trapped among the zinc ions is shown in ball-and-stick representation. **D**- Multiple sequence alignment of Q4Q7F8-F3, as encoded in the *LmjF.30.1460-1510* cluster. The five regions corresponding to homologous active sites in homologous enzymes are highlighted in yellow boxes. **E**- Ecto-3'-nucleotidase activity of 2x10$^6$ intact logPh or statPh *L. major* and secreted enzymes

in the corresponding supernatant (SN) was measured in a colorimetric assay as μM $P_i$ released from 3'-AMP in one hour. n = 4. **F**- Ecto-5'-nucleotidase activity of $2x10^6$ intact logPh or statPh *L. major* and secreted enzymes in the corresponding supernatant (SN) was measured in a colorimetric assay as μM $P_i$ released from 5'-AMP in one hour. n = 3. **G**- Relative endonuclease activity measured by degradation of double-stranded M13 phage genome on $2x10^6$ intact logPh or statPh *L. major* promastigotes. Pellet and supernatant were separated and circular, double-stranded M13 DNA (dsM13) added for 60 min. Remaining intact substrate was assessed on a 0.8% TAE-agarose gel, semi-quantified using ImageJ and normalized to intact substrate not incubated with parasites. n = 3. **H**- Representative gel of **G**, box denotes area of semi-quantification. dsM13: undigested product; *Lm*: control with parasites but no substrate; DNase: Substrate incubated with DNase, first and last lane: DNA ladder.

stimulated with IFNγ exhibit the highest share of cells among tested macrophage phenotypes positive for its cell surface receptors $A_{2a}$ and $A_{2b}$ (Fig BA and BB in S1 Appendix). Activation of these receptors can result in anti-inflammatory immune modulation. Thus, we examined whether macrophage exposure to adenosine or 3'-AMP during *L. major* infection can reduce the pro-inflammatory cytokine response. Indeed, TNFα release from macrophages 3 hours post *Leishmania* infection decreases dose-dependently with increasing concentrations of adenosine. At 100 μM adenosine, TNFα levels align with those of uninfected controls (Fig 3A and Fig BC in S1 Appendix). Addition of 3'-AMP to the infection can mimic the effect observed with adenosine to some extent.

TNFα is a known driver of the antileishmanial host response [51]. Neutralization of TNFα using specific monoclonal antibodies was previously shown to reduce *Leishmania*-induced proliferation of autologous peripheral blood lymphocytes (PBLs) in a hMDM:PBL coculture [52]. T cells are the predominant cell type within PBLs. Thus, we tested whether extracellular adenosine exposure impacts CD14- PBL proliferation 4 days post infection. Although parasite burden of macrophages was not significantly affected, an underlying trend towards an increased parasite burden upon addition of adenosine or 3'-AMP was observed (Fig 3B and Fig BD in S1 Appendix). Relative to the control, reduction in proliferation of PBLs and TNFα secretion indeed was apparent for samples treated with adenosine, 3'-AMP again mimicked this (Fig 3C). Addition of adenosine or 3'-AMP to uninfected cocultures did not affect the share of proliferating PBLs. The proportion of proliferating cells increased greatly upon infection (Fig BE in S1 Appendix). Additionally, a mild to moderate correlation between relative PBL proliferation and TNFα secretion or parasite burden in infected samples was identified (Fig 3D and 3E), suggesting accessory factors induced by adenosine signaling also impact the observed effects. Thus, by generating adenosine through their 3'-nucleotidase activity, *L. major* can reduce certain elements of an early immune response.

**p1/s1 is dispensable for *L. major* but knockout mutants show reduced 3'-nucleotidase/nuclease activity**

To confirm that the observed ecto-3'-nucleotidase/nuclease activity of *L. major* arises from p1/s1, the generation of knockout parasites lacking the *p1/s1* cluster was attempted. However, CRISPR-Cas9-based double homologous replacement approaches [28] failed to yield viable null mutants when sgRNAs binding downstream of *LmjF.30.1460* and upstream of *LmjF.30.1510* were used (Fig 2A). Targeting single genes in the *p1/s1* cluster was not possible due to very high homology between coding and intergenic regions within the cluster. Neither facilitated approaches, using stable genetic integration of a *LmjF.30.1500* construct into the *SSU* locus, or nutritional complementation by supplementing 100 μM adenosine to the medium, yielded viable clones.

Thus, an inducible diCre approach [29] was applied. Excision of *LmjF.30.1460-1510* was induced by addition of rapamycin, that catalyzes dimerization of the Cre recombinase in two clones (E10 and F4) carrying a floxed *p1/s1* cluster (Fig 4A). The eGFP gene of the 5' flox casette is excised together with the gene cluster during the recombination event. Therefore, loss of eGFP signal over time was used to track successful recombination. Up to 7 days post rapamycin induction, eGFP positivity was lost continuously in both clones, while a 3' integrated mCherry signal was retained (Fig 4B and 4C). Following serial dilution to yield single-cell clones, genomic loss of *LmjF.1500/10* and *eGFP* was confirmed by diagnostic PCR (Fig 4D and 4E). One confirmed *L. major*$^{Δp1/s1}$ single-cell clone per floxed strain was chosen for further experiments as induced *p1/s1* knockouts (iKOs). Based on *L. major*$^{Δp1/s1}$ iKO E10, an addback strain was established by genomic

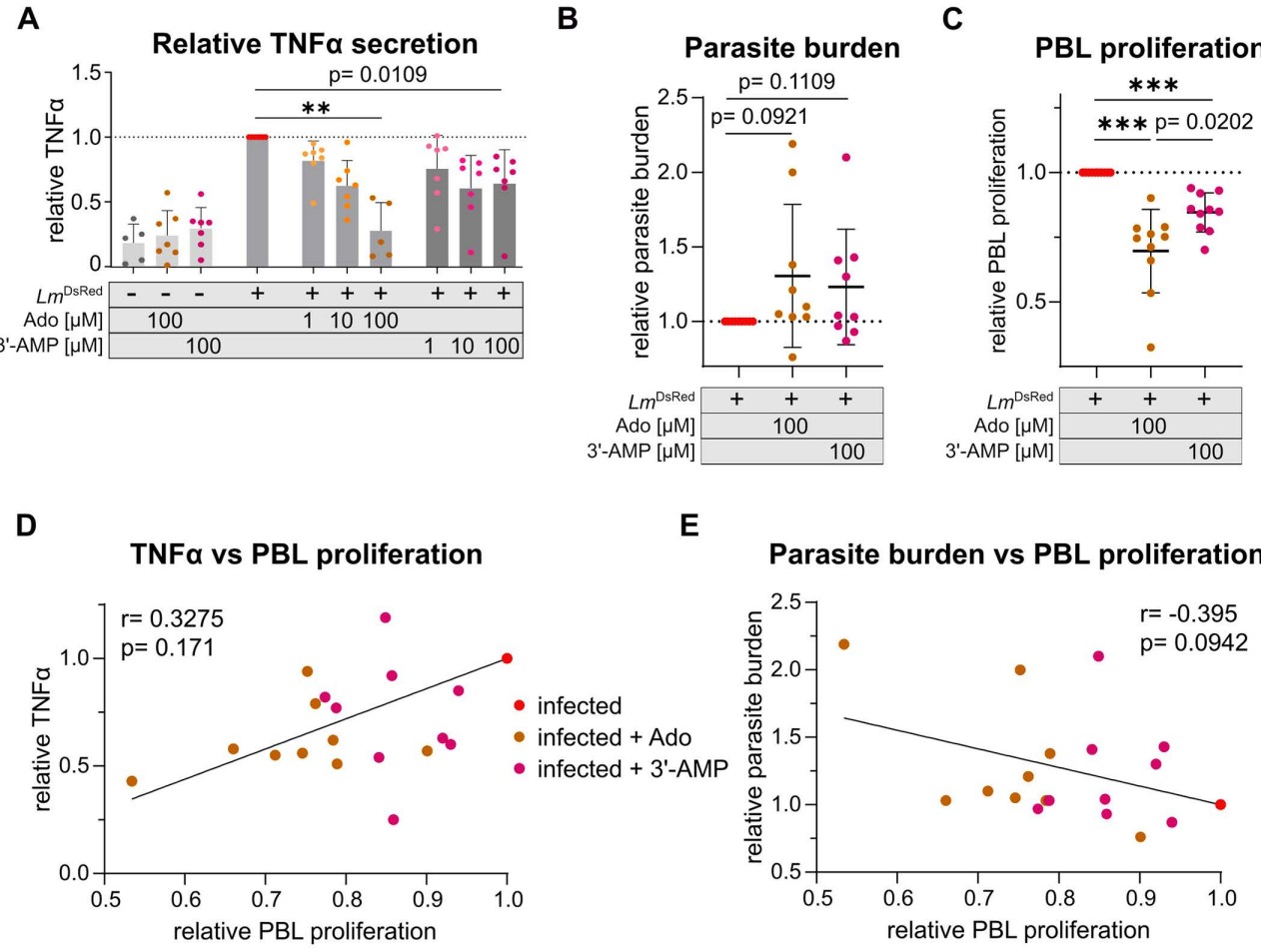

**Fig 3. Presence of adenosine or 3'-AMP during *L. major*<sup>DsRed</sup> infection of hMDM decreases TNFα secretion and PBL proliferation.** hMDM (GM-CSF+IFNγ) were infected with statPh *L. major*<sup>DsRed</sup> promastigotes.3 hours post infection (pi), extracellular parasites were washed off and autologous peripheral blood lymphocytes (PBLs) stained with CellTrace Far Red were added in a 1 macrophage: 5 PBLs ratio. During infection and coculturing, adenosine (Ado) or 3'-AMP were maintained in the indicated concentrations. **A**- Soluble TNFα secreted from hMDM (GM-CSF + IFNγ) was determined in supernatant collected 3 hours post infection with MOI = 10 using ELISA and normalized to the infected sample without additives. n = 5-7 donors in 4 independent experiments. Absolute values available in Fig BC in S1 Appendix. **B**- Relative parasite burden measured in flow cytometry as DsRed mean fluorescence intensity of infected hMDM (GM-CSF + IFNγ) in a hMDM:PBL coculture and normalized to the infected control without additives, determined 4 days pi with a MOI = 20 of *L. major*<sup>DsRed</sup>. Absolute values available in Fig BD in S1 Appendix. **C**- Percentage of proliferated PBLs in a hMDM:PBL coculture was determined as viable CellTrace<sub>low</sub> lymphocytes 4 days pi with a MOI = 20 of *L. major*<sup>DsRed</sup> and normalized to the infected control without additives. Absolute values available in Fig BE in S1 Appendix. **D**- Pearson correlation analysis of relative PBL proliferation against relative TNFα secretion and **E**- relative PBL proliferation and relative parasite burden. r: Pearson correlation coefficient. **B-E**: n = 9-10 donors in 4 independent experiments.

integration of *LmjF.1500* into the *SSU* locus. In comparable culture conditions, both *p1/s1* iKO strains show greatly reduced parasite density in the logPh and statPh compared to the parental strain, whereas the addback strain showed intermediate growth behavior (Fig 4F).

When using *L. major*<sup>Δp1/s1</sup> for 3'-nucleotidase activity assays, both null mutants show considerably diminished activity in the supernatant fraction, whereas the addback partly restored activity. 3'-nucleotidase activities on the pellet fraction are inconclusive but suggest no difference between parental and knockout strains (Fig 5A). Similarly, *L. major*<sup>Δp1/s1</sup> exhibit declined endonuclease activity on double-stranded substrate (Fig 5B and 5C) and delayed

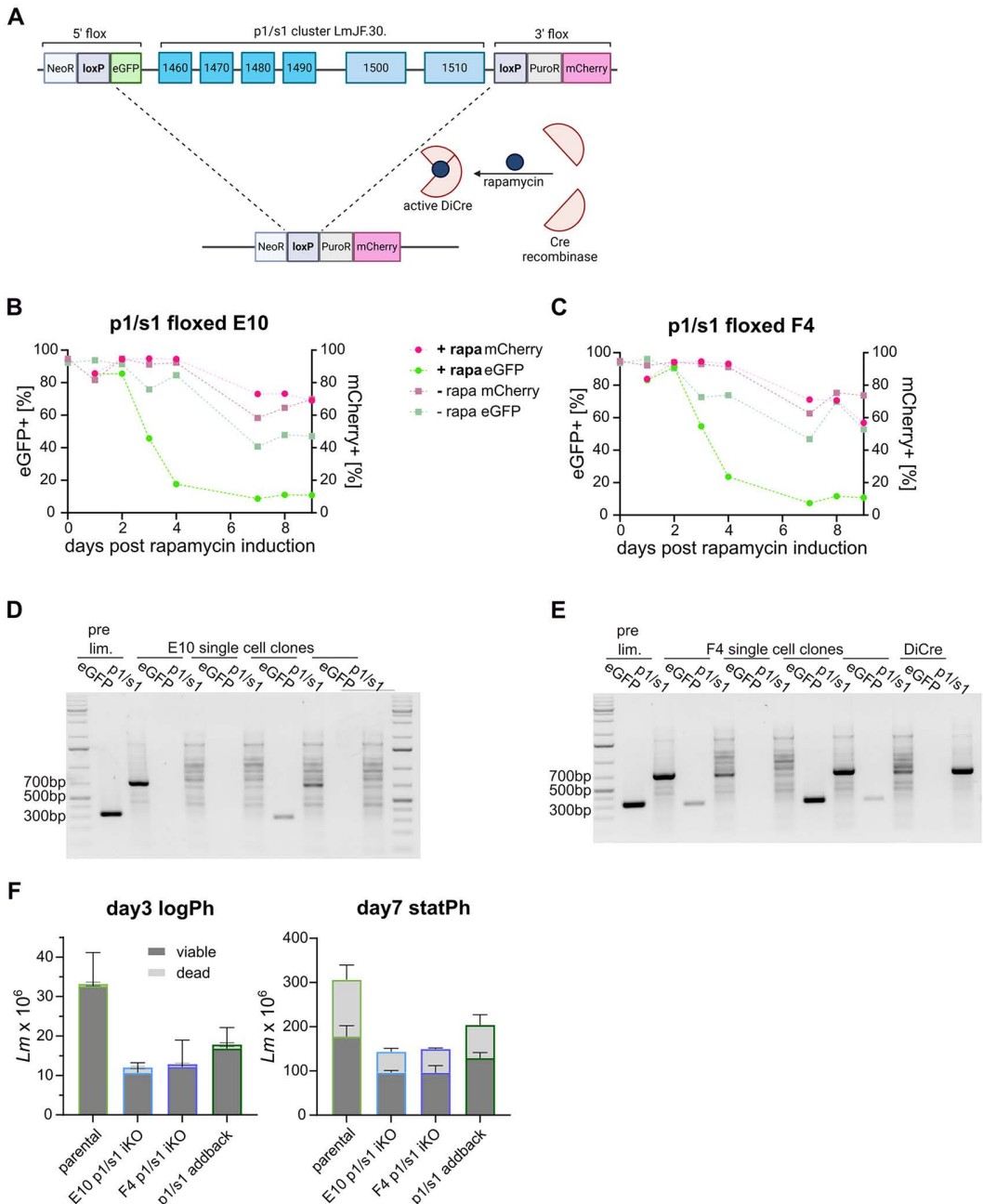

**Fig 4. DiCre based inducible knockout approach yields *L. major p1/s1* null mutants. A**- Schematic visualization of the inducible, dimerizable Cre (diCre) system used to generate *L. major*$^{\Delta p1/s1}$. Full floxed parasites with flox cassettes integrated up- and downstream of the *p1/s1* locus (top) expressing a diCre construct were induced with rapamycin to initiate Cre-mediated excision of the *p1/s1* cluster alongside the *eGFP* gene. Successful recombination (bottom) can be traced by the loss of eGFP signal. Created in BioRender. Schmelzle, **S.** (2026) https://BioRender.com/62mz7co. **B**, **C**- eGFP and mCherry signal (circles) upon diCre induction with rapamycin (+rapa) measured in flow cytometry until 9 days post induction in two *p1/s1* full floxed clones E10 and F4. As controls, respective strains not induced with rapamycin (-rapa, squares) were included. **D**- Diagnostic PCR to detect eGFP and p1/s1 loci in genomic DNA of single cell clones derived from floxed clones E10 and **E**- F4 post rapamycin induction. Confirmed *L. major*$^{\Delta p1/s1}$ clones used for further experiments are marked in gray and further named as E10 and F4 induced knockouts (iKO). *L. major*$^{Cas9/T7/diCre}$ (DiCre) and induced, floxed parasites before limiting dilution (pre.dil.) and were used as controls. **F**- Cell numbers of viable and dead *L. major*$^{Cas9/T7/diCre/3'flox}$, *L. major*$^{\Delta p1/s1}$ clones E10 & F4 and *L. major*$^{p1/s1\ addback}$ were counted on day 3 (logPh) and 7 (statPh) post passaging for viable and dead parasites. All strains were grown under selection pressure of the same antibiotics and were seeded as 0.5x10$^6$ viable parasites on day 0. n = 3 of consecutive passages.

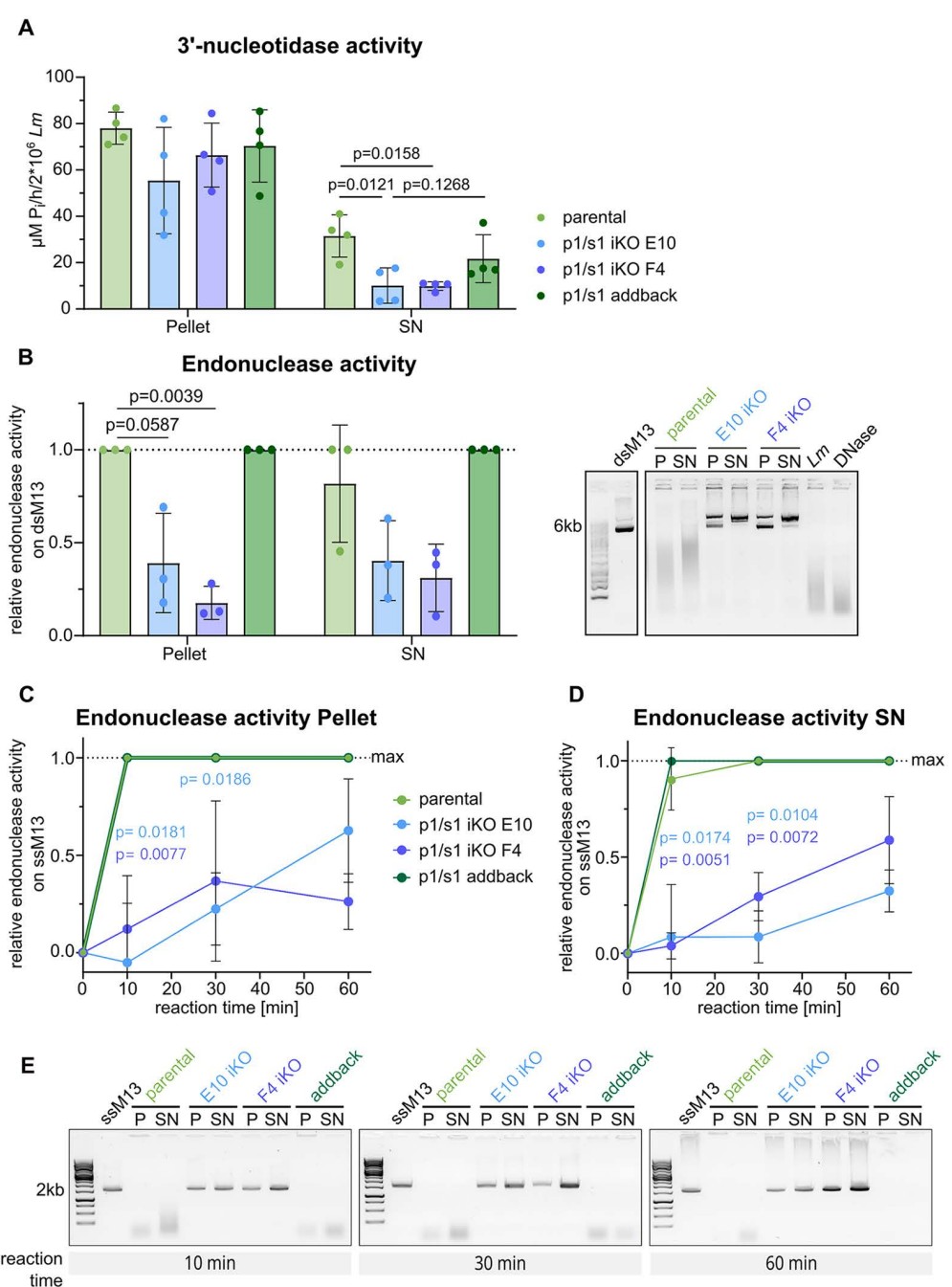

**Fig 5. *L. major p1/s1* null mutants show reduced 3'-nucleotidase/nuclease activity. A-** 3'-nucleotidase activity was measured as µM released $P_i$ from 3'-AMP on $2x10^6$ total, intact *L. major* statPh promastigotes (*L. major*[Cas9/T7/diCre/3'flox] *L. major*[Δp1/s1] clones E10 & F4 and *L. major*[p1/s1 addback]) and the corresponding supernatant (SN). Parasite pellet and supernatant were separated and 3'-AMP as substrate added for 1h, $P_i$ content was determined with a malachite green based assay. n = 4. **B-** Endonuclease activity measured by degradation of double-stranded M13 phage genome on $2x10^6$ total *L. major* statPh promastigotes (*L. major*[Cas9/T7/diCre/3'flox] *L. major*[Δp1/s1] clones E10 & F4 and *L. major*[pSaddback]). Pellet and supernatant (SN) were separated and circular, double-stranded M13 DNA added for 90 min. Remaining intact substrate was assessed on a 0.8% TAE-agarose gel, semi-quantified using ImageJ and normalized to intact substrate not incubated with parasites (dsM13). Representative gel shown. Also included are controls containing only parasites (*Lm*) and dsM13 incubated with DNase. n = 3. Uncropped gel available in Fig C in S1 Appendix. **C, D-** Endonuclease activity on *L. major* pellet and supernatant (SN) fraction was determined as in B, but on single-stranded M13 phage genome (ssM13) for the stated reaction time. n = 3. **E-** Representative gels of C & D.

degradation of single-stranded substrate (Fig 6D-F), both in the pellet and the supernatant fractions. *L. major*$^{\Delta p1/s1}$ addback promastigotes here behaved like the parental strain. Thus, behavior of the parental strain coincided with 3'-nucleotidase/nuclease activity observed in Fig 2, whereas null mutants exhibit a distinct phenotype with delayed or reduced degradation of substrates. The *L. major* genome codes for further 3'-nucleotidases/nucleases besides p1/s1. These redundant enzymes Q66VY6, Q4Q630 and Q4QGQ3 all possess an active site homologous to p1/s1, so residual enzymatic activity in mull mutants was expected.

### *L. major* can compensate for *p1/s1* loss by enrichment of purine salvage proteins

Essentiality of the *p1/s1* cluster remains elusive, as conventional and facilitated CRISPR-Cas9 methods were lethal to the parasites (n = 6) but inducible gene deletion using the same sgRNA templates yielded viable mutants. This observation raised the hypothesis that the gradual depletion of *p1/s1* during diCre induction allowed adaptation of the parasites to compensate for this deletion. To test this, we measured the parasites' 5'-nucleotidase activity. This activity, serving as an alternative source of extracellular adenosine for purine salvage, can aid the parasites in supplying metabolite needs. Indeed, we found 5'-nucleotidase activity to be elevated on *L. major*$^{\Delta p1/s1}$ (Fig 6A).

To quantitatively assess global proteomic dynamics in *p1/s1* null mutants and expression of further purine salvage-linked proteins, a label-free mass spectrometry analysis was applied. In total, 5890 proteins could be detected, corresponding to 73.28% of the reference proteome.

A principal component analysis revealed similarity between replicates of both *p1/s1* iKO strains, whereas the parental strain clustered separately (Fig 6B). Successful depletion of p1/s1 was confirmed at the protein level. We found 604 proteins to be differentially expressed between parental strain and iKO E10, 788 between parental strain and iKO F4, with 467 regulated proteins being shared by both comparisons (Fig DA in S1 Appendix). Focusing only on more abundant proteins in *p1/s1* null mutants, 175 are shared by both deletion strains (Fig 6C). Strikingly, among the proteins with strongly increased abundance in *L. major*$^{\Delta p1/s1}$ strains are enzymes with redundant 3'-nucleotidase-nuclease activity, membrane-bound phosphatases and purine transporters (Fig 6D and 6E; Table 3). Gene ontology terms for purine nucleotide binding, adenyl nucleotide binding, and nucleotide binding were significantly enriched for *p1/s1* iKO E10 but not for *p1/s1* iKO F4 (Fig DB and DC in S1 Appendix). All of these hits can be involved in purine salvage by nucleic acid and nucleotide catabolism or purine precursor transport, providing the parasites with purines despite their reduced 3'-nucleotidase/nuclease activity. Notably, a xanthine phosphoribosyltransferase (Q4QCC2) as key enzyme in intracellular *Leishmania* purine metabolism [53], was more abundant in *p1/s1* iKO E10. Since no copy number variations were detected in the genomic loci of these compared to a parental strain (NCBI SRA data: PRJNA1330523; https://www.ncbi.nlm.nih.gov/sra/PRJNA1330523, Fig E in S1 Appendix), upregulation by genomic adaptation is unlikely. This rather points to a posttranscriptional regulation, as typical for *Leishmania*.

### Loss of p1/s1 does not impact cell death hallmarks upon miltefosine treatment

A 3'-nucleotidase/nuclease in *L. infantum* was shown to play a role in parasite susceptibility to miltefosine, an anti-leishmanial drug [54]. As we have identified p1/s1 as highly abundant in pre-cell death stressed promastigotes, we speculated whether it has a role in parasite cell death, potentially as executer by fragmenting DNA. However, upon treatment with increasing doses of miltefosine, null mutants did not show altered sensitivity compared to the parental or addback strains, neither in viability, phosphatidylserine exposure nor DNA fragmentation (Fig 7A-C). Half-maximal effective concentrations (EC$_{50}$) of miltefosine (Fig 7D) neither varied for the null mutants nor indicated opposed effect for the addback strain. Thus, p1/s1 does not seem to be involved in intracellular cell death pathways like DNA degradation or miltefosine sensitivity of *L. major*. Alternatively, its involvement could be masked by compensation through redundant enzymes.

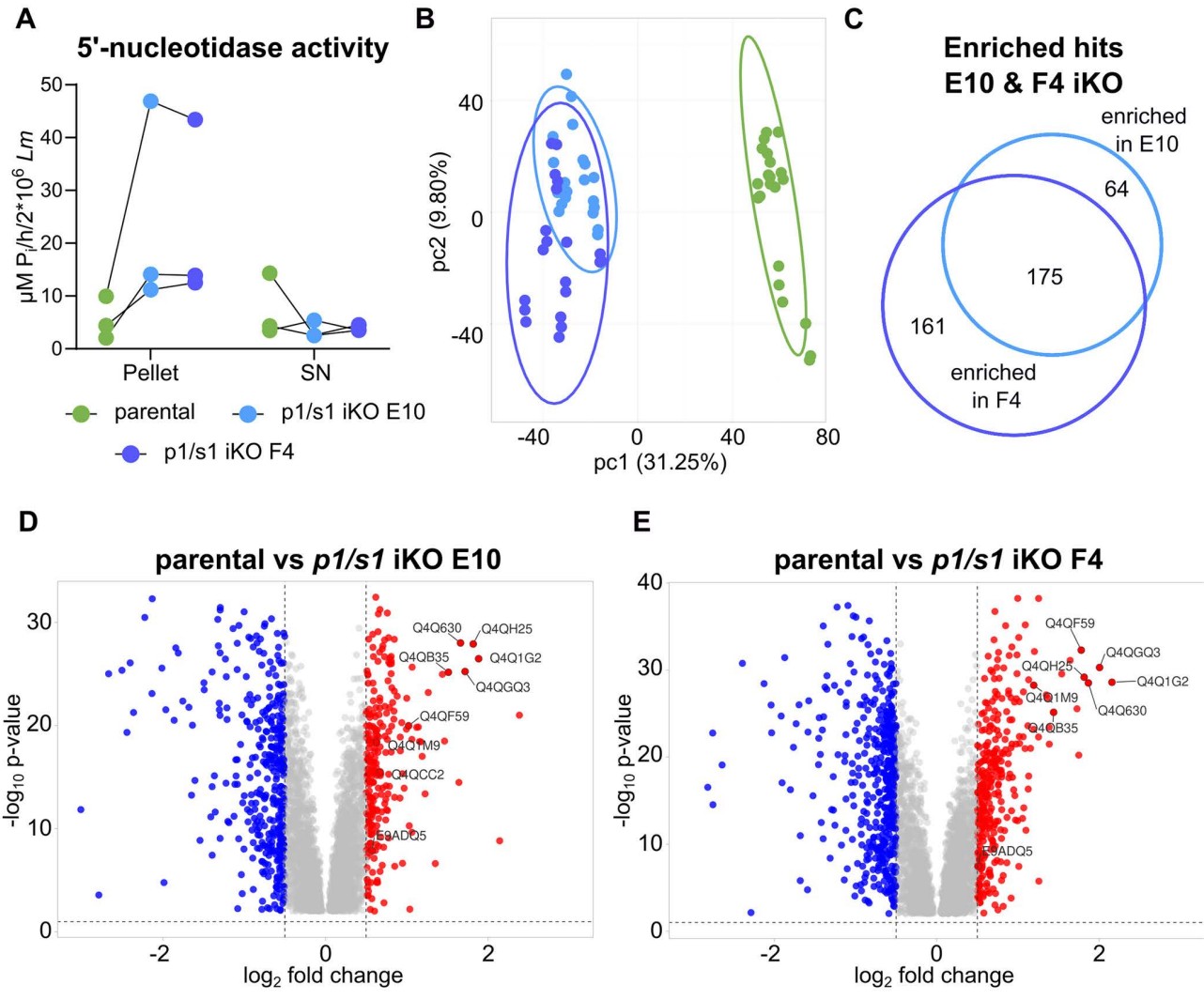

**Fig 6.** ***L. major p1/s1*** **null mutants bypass loss of p1/s1 via redundant purine salvage. A**- Ecto-5'-nucleotidase activity of $2 \times 10^6$ intact statPh *L. major* promastigotes (*L. major*^Cas9/T7/diCre^, *L. major*^Δp1/s1^ clones E10 & F4) and secreted enzymes in the corresponding supernatant (SN) was measured in a colorimetric assay as μM $P_i$ released from 5'-AMP in one hour. n=3. **B**- Principal component analysis of the quantitative proteome data generated on *L. major*^Δp1/s1^ clones E10 & F4 and a parental strain (*L. major*^Cas9/T7/diCre/3'flox^) by label-free mass spectrometry. **C**- Venn diagram showing the number of proteins with increased abundance in *L. major*^Δp1/s1^ clones compared to their parental strain. Template created using [43]. **D, E**- Volcano plot of quantified protein abundance of *L. major*^Δp1/s1^ clones E10 & F4, respectively, in comparison to their parental strain. Cutoffs were set at log$_2$FC<-0.5,>0.5 and p-value<0.01. Denoted are highly abundant proteins of the purine salvage pathway listed in Table 3 Volcano plots visualized with VolcaNoseR [42]. **B-E** Data analysis based on n=7 biological replicates in 3 technical repeats each.

## *p1/s1* null mutants cannot decrease PBL proliferation in hMDM:PBL coculture infection and are more susceptible for neutrophil killing

To determine whether p1/s1 serves as virulence factor during macrophage infection, *L. major*^Δp1/s1^ were used in an *in vitro* infection model. In the promastigote stage, *p1/s1* null mutants showed growth deficiency (Fig 4F) and reduced utilization of extracellular 3'-AMP and nucleic acids (Fig 5). Presuming that both effects are transferable to intracellular amastigotes, they could lead to reduced parasite persistence in host macrophages. However, in an *in vitro* hMDM:PBL coculture

**Table 3. Purine salvage proteins with increased abundance in both L. major^Δp1/s1 clones compared to a parental strain, ranked by p-value in E10 comparison.**

| UniProt Code | Annotation | Log$_2$ ratio E10 | p-value E10 | Log$_2$ ratio F4 | p-value F4 |
|---|---|---|---|---|---|
| Q4Q630 | Putative 3'-nucleotidase/nuclease | 1.662 | $9.88999 \times 10^{-29}$ | 1.868 | $3.02627 \times 10^{-29}$ |
| Q4QH25 | Nucleobase transporter | 1.819 | $1.24394 \times 10^{-28}$ | 1.816 | $6.61783 \times 10^{-30}$ |
| Q4Q1G2 | Putative membrane-bound acid phosphatase 2 | 1.886 | $3.33012 \times 10^{-27}$ | 2.159 | $2.5267 \times 10^{-29}$ |
| Q4QGQ3 | Putative 3'-nucleotidase/nuclease | 1.719 | $5.82386 \times 10^{-26}$ | 2.006 | $5.09025 \times 10^{-31}$ |
| Q4QB35 | Membrane-bound acid phosphatase 2 | 1.514 | $6.83801 \times 10^{-26}$ | 1.44 | $7.14518 \times 10^{-26}$ |
| Q4QF59; Q4QF58 | Putative nucleoside transporter 1 | 1.023 | $1.05811 \times 10^{-20}$ | 1.78 | $5.1277 \times 10^{-33}$ |
| Q4Q1M9 | Inosine-guanosine transporter | 0.63 | $3.89054 \times 10^{-19}$ | 1.195 | $5.58245 \times 10^{-29}$ |
| E9ADQ5 | Exonuclease domain-containing protein | 0.575 | $1.37787 \times 10^{-08}$ | 0.508 | $1.23532 \times 10^{-07}$ |

system, infectivity of *L. major*^Δp1/s1 compared to the parental strain neither decreased in terms of infection rate nor parasite burden (Fig 8A and 8B). This also confirmed that the slightly decreased proportion of metacyclic promastigotes in the p1/s1 iKO (Fig FA in S1 Appendix) does not to affect macrophage infection. As shown above, we observed a reduction in PBL proliferation when adenosine or 3'-AMP was added to a hMDM:PBL coculture during infection (Fig 3C). *p1/s1* deletion alone did not influence absolute PBL proliferation (Fig 8C). A suppressive effect comparable to Fig 3C was apparent for the parental strain in the presence of adenosine or 3'-AMP. However, *L. major*^Δp1/s1 lost the ability to suppress PBL proliferation when 3'-AMP was added (Fig 8D), likely due to reduced degradation to adenosine. Addition of adenosine as a positive control still reproduced this reduction.

Besides the p1/s1 3'-nucleotidase activity that potentially counteracts inflammatory signaling by generating extracellular adenosine, the endonuclease activity of p1/s1 can also bring advantages for parasite survival in mammalian hosts. Ecto-nuclease activity of p1/s1 can be beneficial for *Leishmania* promastigotes when encountering neutrophils in the early stages of infection. In response to pathogen detection, neutrophils can extrude neutrophil extracellular traps (NETs) that have anti-microbial capacity. Several pathogens possess extracellular nucleases to evade NET-mediated killing [55–58]. *L. major* was shown to induce NET formation upon neutrophil infection [59, 60]. Whether p1/s1 allows *L. major* to degrade and survive NETs was examined on human primary neutrophils (PMNs) and cell-free NET-enriched supernatants. Control promastigotes of the parental strain were able to efficiently degrade NET DNA, whereas *L. major*^Δp1/s1 were less able to do so (Fig 8E and 8F). This effect was prominent for both parasite pellet and supernatant, comparable to nuclease activity shown in Fig 5B. To test for antileishmanial effects of NETs on parental and p1/s1 deficient strains, parasites were incubated with PMNs whose phagocytosis was inhibited by Cytochalasin D. The majority of parental strain promastigotes survived this co-incubation but both *L. major*^Δp1/s1 strains were significantly reduced in number of surviving parasites. When we added DNaseI to degrade NETs, these leishmanicidal effects were lost and survival of the iKO strains resembled that of the parental strain (Fig 8G).

Taken together, both of the dual p1/s1 activities are able to support evasion of different branches of the innate immune response. Thus, this ecto-enzyme is key in host-*Leishmania* interactions during early infection.

## Discussion

3'-nucleotidase/nuclease activities of several trypanosomatid species have been biochemically characterized and described as ecto-enzymes with manifold functions before [3,12,61,62]. In the present study, we combined quantitative proteomic and genomic editing approaches in *L. major* with human primary *in vitro* infection models to advance knowledge about the importance of p1/s1 in early infection. As enzyme with a dual activity, we were able to link both functions to critical metabolite acquisition as well as to immune escape mechanisms used by the parasite to promote pathogenicity. Furthermore, we utilized quantitative proteomics to dissect how *L. major* adapts to p1/s1 deletion by enriching complementary purine salvage proteins. This highlights the importance of p1/s1 for parasite survival under stress conditions.

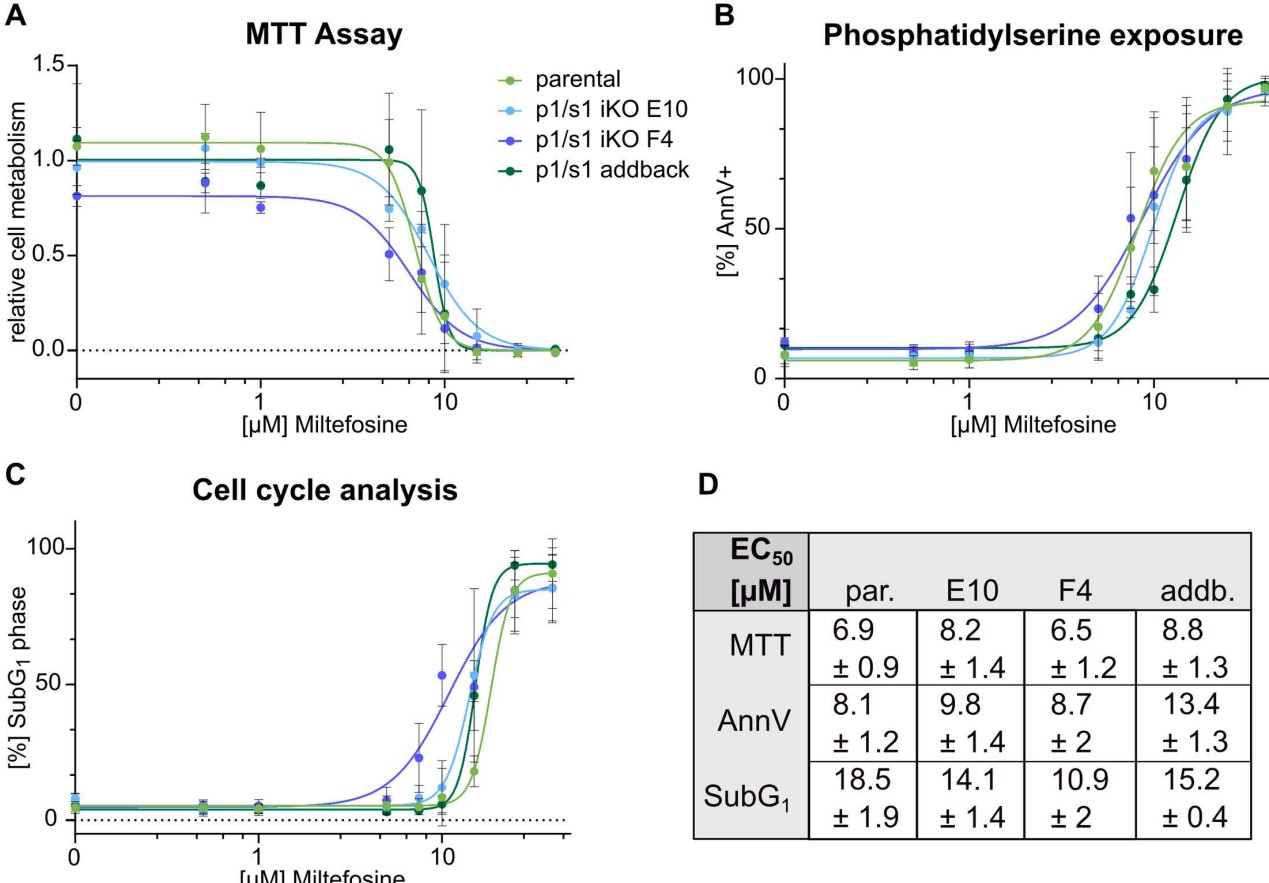

**Fig 7. *L. major p1/s1* null mutants do not show altered cell death hallmarks upon miltefosine treatment.** *L. major* logPh promastigotes (*L. major*-Cas9/T7/diCre/3'flox, *L. major*Δp1/s1 clones E10 & F4 and *L. major*p1/s1 addback) were treated with increasing concentrations of miltefosine (0.1-60μM) for 48 h and cell death hallmarks analyzed by different assays. **A**- Cellular metabolism of 5x10^6 parasites was quantified using a colorimetric MTT assay, absorbance at 540 nm was normalized to the sample containing 1.5% DMSO as carrier control; n = 3. **B**- Phosphatidylserine exposure on the cell surface of 2x10^6 parasites was stained using FITC labelled AnnexinV. Percentage of phosphatidylserine positive parasites (AnnV+) was quantified by flow cytometry; n = 3–5. **C**- 2x10^6 parasites were analyzed for fragmented DNA, as seen in the SubG$_1$ phase, by stochiometric DNA staining using SYTOXGreen analyzed by flow cytometry; n = 3–5. **D**- Summarized EC$_{50}$ values obtained from non-linear 4-parameter regression of all cell death assays for *L. major*Cas9/T7/diCre/3'flox (par.), *L. major*Δp1/s1 clones E10 & F4 and *L. major*p1/s1 addback (addb.); ± 95% confidence interval.

Proteomic analysis revealed that p1/s1 is highly abundant in both drug-induced stress conditions that would eventually lead to cell death at longer incubation. p1/s1 is also more abundant in statPh promastigotes, suggesting that its expression is both stage-specific and inducible. This observation supports the idea that 3'-nucleotidases/nucleases can be induced not only by nutrient depletion [63], but also as part of a stress adaptation or survival mechanism prior to cell death. Possibly, its involvement in nucleotide and nucleic acid catabolism aids the parasites in acquiring essential purine salvage precursors, as 3'-nucleotidase is the dominant ecto-nucleotidase activity in *L. major*. p1/s1, due to its N-terminal signal sequence, was found to accumulate as secreted enzyme in the supernatant of a promastigote culture. No protein was detected in the parasite lysate, likely due to immediate secretion after protein biosynthesis.

Recent attention has focused on a 3'-nucleotidase/nuclease as genetic marker for miltefosine resistance in *L. infantum* [54]. However, we did not observe evidence for a direct involvement of p1/s1 in cell death, neither regarding sensitivity of

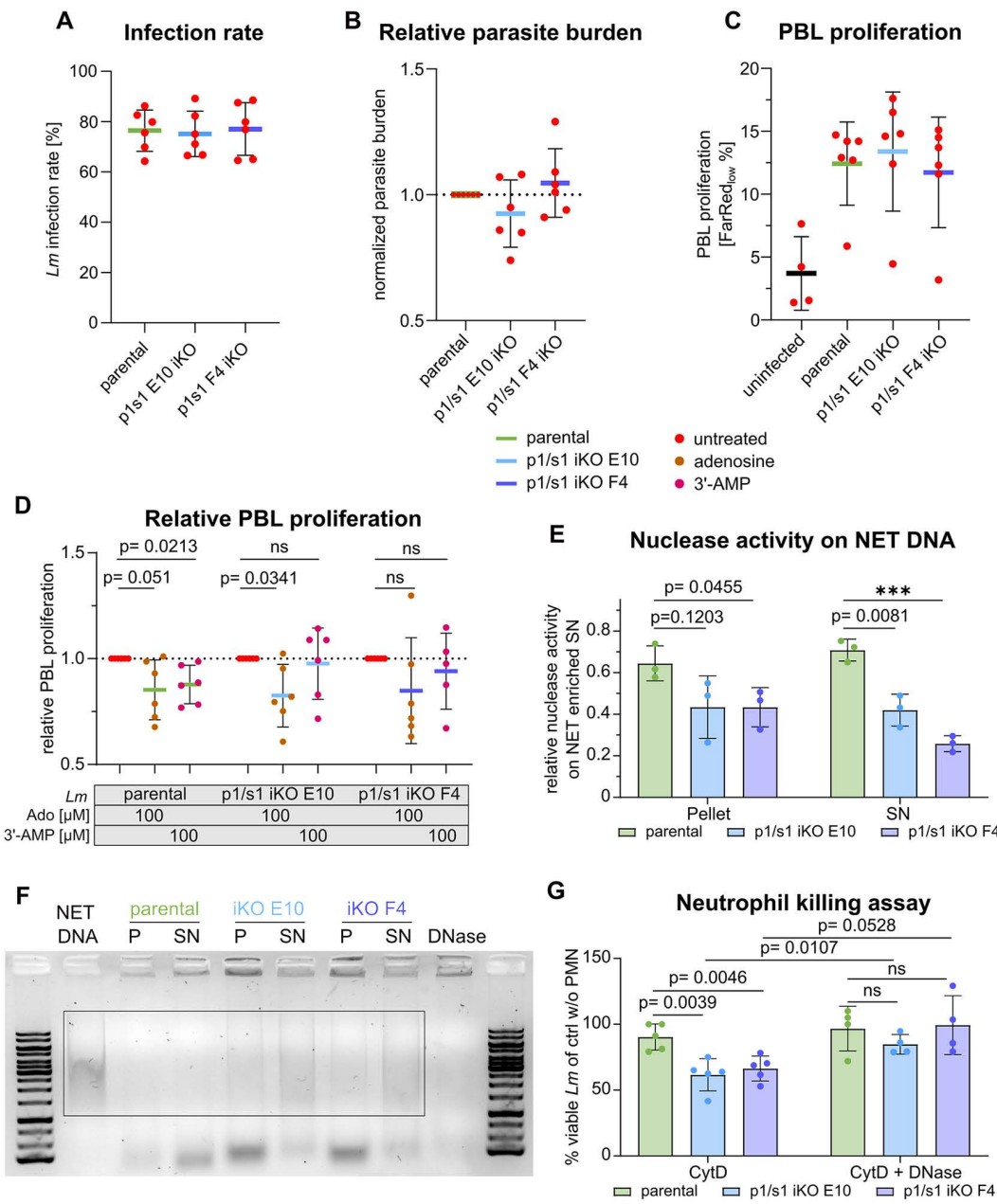

**Fig 8. *L. major p1/s1* null mutants cannot decrease PBL proliferation upon presence of 3'-AMP and show decreased degradation of neutrophil extracellular traps.** hMDM (GM-CSF + IFNγ) were infected with statPh *L. major*Cas9/T7/diCre/3'flox, *L. major*Δp1/s1 clones E10 or F4 promastigotes at an MOI=20. 3 hours post infection (pi), extracellular parasites were washed off and autologous peripheral blood lymphocytes (PBLs) stained with CellTrace Far Red were added in a 1 macrophage: 5 PBLs ratio for 4 days. During infection and coculturing, adenosine (Ado) or 3'-AMP were maintained at 100 μM, if applicable. Intracellular *L. major* were stained with α-*Lm* mouse serum and labelled with α-mouse-AlexaFluor647 antibody. **A**- *L. major* infection rate of macrophages in a hMDM:PBL coculture 4 days pi. The percentage of AlexaFluor647+ viable macrophages was determined in flow cytometry to give the share of infected macrophages. **B**- Parasite burden 4 days pi in a hMDM:PBL coculture. Parasite burden was measured as AlexaFluor647 mean fluorescence intensity of viable, infected macrophages and normalized to the control infected with parental strain. **C**- Percentage of proliferated PBLs in a hMDM:PBL coculture was determined as viable CellTrace$_{low}$ lymphocytes 4 days pi. **D**- Normalized share of proliferated PBLs in a hMDM:PBL coculture 4 days pi, relative to the respective strain's infected control without adenosine or 3'-AMP. Absolute percentage of PBL proliferation is provided in Fig FB in S1 Appendix. **A**-**D**: n = 6 donors in 3 independent experiments, Welsh's t-test. Human primary neutrophils were used to study p1/s1 dependent degradation of neutrophil extracellular trap (NET) degradation. **E**- Nuclease activity measured by degradation of NET enriched supernatants on 2x10⁶ intact *L. major* statPh promastigotes (P, *L. major*Cas9/T7/diCre/3'flox *L. major*Δp1/s1 clones E10 & F4) and the respective supernatant (SN). Pellet and supernatant

were separated and 500µg NET DNA added for 2 hours. Remaining intact NET DNA scaffold was assessed on a 0.8% TAE-agarose gel, semi-quantified using ImageJ and normalized to substrate not incubated with parasites. n = 3 donors in three independent experiments. **F**- Representative gel of **E**, box indicates area used for semi-quantification with ImageJ. **G**- *L. major* statPh promastigotes (*L. major*$^{Cas9/T7/diCre/3'flox}$ *L. major*$^{\Delta p1/s1}$ clones E10 & F4) were incubated with human neutrophils at MOI = 1, optionally in the presence of DNase, for 3 hours at 37 °C. CytochalasinD (CytD) inhibits phagocytosis of parasites into neutrophils. After further incubation at 27 °C for 2 days, motile, viable promastigotes were counted for each sample and normalized to the respective control incubated without neutrophils. n = 4–5 donors in 2–3 independent experiments.

deletion strains towards miltefosine, nor intracellular DNA degradation or phosphatidylserine exposure upon treatment. Thus, a link between miltefosine sensitivity and 3'-nucleotidases/nucleases might be strain- or protein cluster-specific.

Essentiality of p1/s1 remains unclear since direct or facilitated knockout approaches did not yield viable parasites. In contrast, an inducible deletion method yielded two viable Δp1/s1 strains, despite the application of identical sgRNA and recombination sequences as in the aforementioned approaches. However, reduced growth of the deletion strains possibly originates from insufficient nutrient acquisition from the culture medium, emphasizing *L. major* dependency on purine uptake. Genome plasticity in *Leishmania* can lead to rapid adaptation in response to *p1/s1* loss during diCre induction, which is not achieved during direct knockout generation [64]. Indeed, comparing quantified proteomes revealed dynamic changes in protein expression. Elevated levels of proteins associated with purine salvage, transport, and redundant enzymes were detected in both Δp1/s1 strains, similar to the translational adaptation and increased adenosine uptake seen in parasites under purine starvation conditions [65–67]. In fact, we found no evidence for genomic rearrangement in terms of copy number variations of affected genes in the genome of *L. major*$^{\Delta p1/s1}$ strains, supporting a genome-independent adaptation. These findings can explain how *L. major*$^{\Delta p1/s1}$ circumvent essentiality of p1/s1 by post-translation compensation strategies to eventually meet their nutrient requirements.

*L. major* lacking the *p1/s1* cluster showed reduced 3'-AMP and nuclease degradation, confirming that both activities directly derive from p1/s1. Differential iKO effects for 3'-nucleotidase and nuclease on the parasite pellet could be linked to *LmjF.30.1460-1490*, predicted to be membrane-bound, possessing only nuclease activity. These class I nucleases might only participate in extracellular nucleic acid degradation for nutrient acquisition and NET escape. As expected from literature, substrate preference for ssDNA over dsDNA was apparent, as single-stranded substrate is digested in a much shorter time [3]. Residual enzymatic activity arises from redundant 3'-nucleotidases/nucleases still present in the parasite genome, which are increased on the protein level.

Besides the predominant expression of the *p1/s1* cluster in the statPh, *L. major* possesses three additional putative 3'-nucleotidases/nucleases, *LmjF.12.0400* and *LmjF.31.2300-10*. The two detected in our proteomics data set also tend to be more abundant in the statPh. Accordingly, overall 3'-nucleotidase/nuclease activity is higher in statPh than in logPh promastigotes. Enzymatic activity in the supernatant fraction derives from secreted proteins and thus corroborates extracellular localization of p1/s1 [15]. Its secretion is likely caused by an N-terminal signal sequence. These findings align with earlier insights on the *L. donovani* p1/s1 homologue *Ld*Nuc$^S$ [8].

Nuclease activity of p1/s1 is not only responsible for degradation of NETs but also allows parasites to escape from them, supporting earlier findings [20]. As published before, NETs do not have leishmanicidal effects on *L. major* parasites, although they are capable of inducing NET formation [58, 59]. However, *L. major*$^{\Delta p1/s1}$ partially lose the ability to withstand NET-induced killing. Reversion of this effect by addition of DNaseI suggests that this is indeed based on reduced ecto-nuclease activity. Thereby, p1/s1 exhibits NET escape capabilities, as also observed for a *L. infantum* 3'-nucleo-tidase/nuclease [19].

Ecto-nucleotidase activity of *Leishmania* has a well-established influence on parasite infectivity in mice, suggesting that the generation of extracellular adenosine impacts disease development [37,68–70]. This hypothesis is supported by human visceral leishmaniasis patient data, as well as the finding of immunosuppressive effects of adenosine in *Phlebotomus* saliva [71, 72].

In our human primary *in vitro* infection setting, addition of adenosine or 3'-AMP can enhance parasite burden in human macrophages. At the same time, both additives decrease immediate secretion of the strongly pro-inflammatory cytokine TNFα. Previously, we showed that TNFα regulates T cell proliferation in response to *L. major* infection and indeed, addition of adenosine or 3'-AMP to a hMDM-PBL coculture also reduced PBL proliferation [52]. As 3'-AMP is hydrolyzed to adenosine by ecto-3'-nucleotidase activity of *L. major*, these effects directly derive from p1/s1 activity. Accordingly, Δ*p1/s1* mutants lose the ability to reduce PBL proliferation in the presence of 3'-AMP.

Taking all findings together, p1/s1 is a multifaceted enzyme involved on several levels during the *L. major* lifecycle. Its dual ecto-activities as 3'-nucleotidase and endonuclease are crucial for nutrient acquisition from hosts, laying the foundations for essential purine salvage. Moreover, p1/s1 is also involved in a vast network of host-parasite interactions, manipulating and shaping innate immune responses during early stages of infection in favor of the parasites. Generation of extracellular adenosine through cleavage of 3'-AMP leads to anti-inflammatory macrophage responses before promastigotes are phagocytosed. Antimicrobial effects of NETs can be circumvented by nuclease activity. However, recent results showed that *L. infantum* with a deletion phenotype for 3'-nucleotidase/nuclease genes have advantages in metacyclogenesis and sand fly transmissibility, marking these enzymes as a double-edged sword [73]. Potentially, the consequences and implications of 3'-nucleotidase/nuclease activity are changing during the *Leishmania* lifecycle, with altered effects in different host environments.

Effects observed in iKO strains are limited due to compensatory capacities of the parasites. Hence, a specific pan-3'-nucleotidase/nuclease inhibitor can have a great potential for further studies on the role of these enzymes for parasite survival and infectivity. Several characteristics of p1/s1 and 3'-nucleotidases/nucleases in general make them interesting as potential drug target: The type I nuclease class is expressed in the infective stages of many protozoa and it has no human equivalent, reducing host toxicity of potential inhibitors [3]. As ecto-enzymes, they are accessible for drug binding and have multiple functions on parasite and host level. Class I nucleases were also shown to be expressed in *Leishmania* amastigotes [8, 14]. Consistently, p1/s1 expression in *L. major* amastigotes is also evident from our proteomics data set. Whether a prompt inhibition is sufficient for *Leishmania* killing or prone to resistance emergence, needs to be studied. Development of a specific inhibitor can be key to determine the role of 3'-nucleotidases/nuclease in established infection as well as studying mechanisms of transmission benefits seen in deletion strains [73]. To accelerate development of such an inhibitor, we are now screening for compounds binding into the predicted active site of p1/s1.

By presenting a comprehensive study on p1/s1, a bifunctional 3'-nucleotidase/nuclease in *L. major*, we elucidated the central role of this ecto-enzyme to emphasize its manifold involvements during infection onset. Our findings highlight the potential of class I nucleases as drug targets, a potential that could be further realized by elucidating the function of p1/s1 in intracellular amastigotes. Overall, 3'-nucleotidases/nucleases in *Leishmania* are highly relevant to study in many aspects along the lifecycle and are key to understand molecular dynamics of *Leishmania* infection.

## Supporting information

**S1 Appendix. Fig A – Cell-death induced in *L. major* promastigotes.** Phosphatidylserine exposure as cell death marker determined in flow cytometry by AnnexinV positivity (AnnV+) of promastigotes treated with 25 µM miltefosine or 25 µM staurosporine for the indicated time, compared to untreated promastigotes derived from the logPh or statPh. n = 3–6. Fig B – Effect of extracellular adenosine generated from 3'-AMP hydrolysis on hMDM infection. A, B- $A_{2a}$ and $A_{2b}$ positivity, respectively, of hMDM with different differentiations and stimulations, as specified. The surface markers were stained with respective specific and labeled antibodies for quantification in flow cytometry. Differentiation stimuli (50 ng/ml or 30 ng/ml GM-CSF) were applied for 5–7 days, followed by 24 h of activation with 50 ng/ml IFNγ or 100 ng/ml LPS, if applicable. n = 3–6 donors in 2–3 independent experiments. C- Absolute values to Fig 3A. hMDM (GM-CSF + IFNγ) were

infected with MOI = 10 statPh *L. major*<sup>DsRed</sup> promastigotes for 3 hours. During infection, adenosine (Ado) or 3'-AMP were maintained in the indicated concentrations. Soluble TNFα secreted from macrophages was determined in supernatant collected 3 hours post infection using ELISA. n = 7 donors in 4 independent experiments. D- Absolute values to Fig 3B. Relative parasite burden measured as DsRed MFI of infected hMDM (GM-CSF + IFNγ) normalized to the infected control without additives. Macrophages were infected for 3 hours with a MOI = 20 of statPh *L. major*<sup>DsRed</sup> in the presence of adenosine or 3'-AMP before autologous PBLs were added to the infection and incubated for 4 days. n=9-10 donors in 4 independent experiments. E- Absolute values to Fig 3C. hMDM (GM-CSF+IFNγ) in co-culture with autologous PBLs were infected with *L. major*<sup>DsRed</sup>. Percentage of proliferated PBLs was determined as viable CellTrace<sub>low</sub> lymphocytes 4 days pi. n=9-10 donors in 4 independent experiments. Fig C – *L. major p1/s1* null mutant endonuclease activity. Uncropped gel image to Fig 5B. Endonuclease activity measured by degradation of double-stranded M13 phage genome on 2x10⁶ total *L. major* statPh promastigotes (*L. major*<sup>Cas9/T7/diCre/3'flox</sup> *L. major*<sup>Δp1/s1</sup> clones E10 & F4 and *L. major*<sup>p1/s1 addback</sup>). Pellet (P) and supernatant (SN) were separated and circular, double-stranded M13 DNA added for 60 min. Remaining intact substrate was assessed on a 0.8% TAE-agarose gel, semi-quantified using ImageJ and normalized to intact substrate not incubated with parasites (dsM13). Representative, uncropped gel shown. Also included are controls containing only parasites (*Lm*) and dsM13 incubated with DNase. Fig D – Comparative proteomics of *L. major p1/s1* null mutants to parental strain. A- Differentially detected proteins and intersections between parental strain *L. major*<sup>Cas9/T7/diCre/3'flox</sup> and *L. major*<sup>Δp1/s1</sup> clones E10 & F4 in a quantitative proteomics analysis. B- Gene ontology enrichment of molecular functions for proteins with increased abundance in Δ*p1/s1* iKO E10 compared to the parental strain. C- Gene ontology enrichment of molecular functions for proteins with increased abundance in Δ*p1/s1* iKO F4 compared to the parental strain. Analysis based on 7 biological replicates and 3 technical replicates each. Fig E – Mapped WGS reads of *L. major*<sup>Δp1/s1</sup> and a parental strain at loci of interest. Mapped whole genome sequencing (WGS) reads of a parental strain and *p1/s1* iKOs (*L. major*<sup>Cas9/T7/diCre/3'flox</sup> *L. major*<sup>Δp1/s1</sup> clones E10 & F4) at loci of interest as detailed in the figure. Loci of interest were identified by elevated protein levels in a comparative proteomics screen (Table 3). Fig F – *L. major p1/s1* null mutants induce comparable PBL proliferation but fail to suppress proliferation with 3'-AMP as the parental strain. A- Proportion of metacyclic promastigotes within the infectious stationary phase of 2x10⁸ parental (day7) or p1/s1 iKO (day 9) parasites as determined by peanut lectin purification. B- Absolute values to Fig 8D. Macrophages were infected for 3 hours with MOI = 20 of statPh promastigotes (*L. major*<sup>Cas9/T7/diCre/3'flox</sup>, *L. major*<sup>Δp1/s1</sup> clones E10 & F4) in the presence of adenosine or 3'-AMP before autologous CD14⁻ peripheral blood lymphocytes (PBLs) were added and incubated for 4 days. Percentage of proliferated PBLs was determined as CellTrace<sub>low</sub>. n = 6 donors in 3 independent experiments. Table A: Data to Fig 3A - Soluble TNFα, normalized to infected, untreated. Table B: Raw data to Fig BC in S1 Appendix - Soluble TNFα [pg/ml]. Table C: Data to Fig 3B – Parasite burden, normalized to untreated. Table D: Raw data to Fig 3B and Fig BD in S1 Appendix – Parasite burden (DsRed MFI) [AU]. Table E: Data to Fig 3C – PBL proliferation, normalized to untreated. Table F: Raw data to Fig BE in S1 Appendix – PBL proliferation (CellTrace<sub>low</sub>) [%]. Table G: Supplemental data to Fig 7 - regression analyses. Table H: Raw data to Fig 8A – Infection rate (DsRed-positive) [%]. Table I: Data to Fig 8B – Parasite burden, normalized to untreated. Table J: Raw data to Fig 8C and Fig FB in S1 Appendix – PBL proliferation (CellTrace<sub>low</sub>) [%]. Table K: Data to Fig 8D – PBL proliferation, normalized to untreated. Table L: Data to Fig 8G – Viable L. major from neutrophil killing assay, normalized to respective control without neutrophils. Supplemental methods: Determination of metacyclic promastigotes. Whole genome sequencing. NGS read processing. Copy number variation analysis. Table M: Primer sequences for whole genome sequencing. Homology modelling of p1/s1.
(DOCX)

**S1 Dataset. Proteome data of cell death-induced L. major used for quantitative analysis as shown in Fig 1.** Contains MaxLFQ values as well as statistical analysis of different condition comparisons. Pro w/o: log phase promastigotes, untreated; Pro SSP6 and Pro SSP24: log phase promastigotes treated with stauroporine for 6 h or 24 h, respectively;

Milte 3 and Milte 24: log phase promastigotes treated with miltefosine for 3 h or 24 h, respectively; Pro 7d: stat phase promastigotes. 7 biological replicates per condition, each measured in 3 technical replicates.
(XLSX)

**S2 Dataset. Proteome data of p1/s1 deletion mutants used for quantitative analysis as shown in Fig 6.** Contains MaxLFQ values as well as statistical analysis of different condition comparisons. L. major Cas9-T7-DiCre (parental), L. major Cas9-T7-DiCre p1s1 iKO F4, L. major Cas9-T7-DiCre p1s1 iKO E10. 7 biological replicates per condition, each measured in 3 technical replicates.
(XLSX)

## Acknowledgments

We thank M-S. Philipp for excellent technical support for flow cytometry experiments and T. Bauer for providing PMNs. Generative AI was only used to enhance comprehensibility and language flow, authors take full responsibility of the content.

## Author contributions

**Conceptualization:** Stella M. Schmelzle, Michaela Bergmann, Peter Kolb, Stefan Tenzer, Katrin Bagola, Ger van Zandbergen.

**Data curation:** Stella M. Schmelzle, Michaela Bergmann, Tanja Ziesmann, Ute Distler, Csaba Miskey, Liam Childs.

**Formal analysis:** Stella M. Schmelzle, Michaela Bergmann, Tanja Ziesmann, Ute Distler, Csaba Miskey, Liam Childs, Ger van Zandbergen.

**Funding acquisition:** Liam Childs, Peter Kolb, Stefan Tenzer, Ger van Zandbergen.

**Investigation:** Stella M. Schmelzle, Michaela Bergmann, Bianca Walber, Jamal Shamsara, Tanja Ziesmann, Ute Distler, Csaba Miskey, Liam Childs.

**Methodology:** Stella M. Schmelzle, Michaela Bergmann, Katrin Bagola.

**Project administration:** Katrin Bagola, Ger van Zandbergen.

**Resources:** Ger van Zandbergen.

**Software:** Tanja Ziesmann, Ute Distler, Csaba Miskey, Liam Childs.

**Supervision:** Katrin Bagola, Ger van Zandbergen.

**Validation:** Stella M. Schmelzle, Katrin Bagola, Ger van Zandbergen.

**Visualization:** Stella M. Schmelzle, Jamal Shamsara, Tanja Ziesmann, Ute Distler, Liam Childs.

**Writing – original draft:** Stella M. Schmelzle.

**Writing – review & editing:** Peter Kolb, Stefan Tenzer, Katrin Bagola, Ger van Zandbergen.

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
