## [Decision Letter · Decision Letter 0]

11 Mar 2026

PPATHOGENS-D-26-00464

p1/s1, a 3'-nucleotidase/nuclease, allows Leishmania major to circumvent host innate immune response mechanisms

PLOS Pathogens

Dear Dr. van Zandbergen,

Thank you for submitting your manuscript to PLOS Pathogens. After careful consideration, we feel that it has merit but does not fully meet PLOS Pathogens's publication criteria as it currently stands. Therefore, we invite you to submit a revised version of the manuscript that addresses the points raised during the review process.

We look forward to receiving your revised manuscript.

Kind regards,

Dominique Soldati-Favre

Section Editor

PLOS Pathogens

Sumita Bhaduri-McIntosh

Editor-in-Chief

PLOS Pathogens

orcid.org/0000-0003-2946-9497

Michael Malim

Editor-in-Chief

PLOS Pathogens

orcid.org/0000-0002-7699-2064

**Additional Editor Comments:**

Based on the reviewers’ reports, the manuscript requires minor revisions before it can be accepted for publication.

**Journal Requirements:**

- ® on page: 9

- TM on page: 8.

Potential Copyright Issues:

i) We note that Figures 2A, and 4A are created through BioRender. Please confirm that you hold a Premium account and provide a pdf copy of the CC BY 4.0 Licence as provided by BioRender. For instructions on how to generate a CC BY 4.0 license for your figure, please see the guidelines here: https://help.biorender.com/hc/en-gb/articles/21282341238045-Publishing-in-open-access-resources.

If you are using the free assets from BioRender, we are unable to publish these images as they are licenced under a stricter licence than CC BY 4.0. In this case we ask you to remove the BioRender images and replace them with open source alternatives.

See these open source resources you may use to replace images / clip-art:

- https://bioart.niaid.nih.gov/

- https://bioicons.com/

- https://healthicons.org/

- https://scidraw.io/

- https://reactome.org/icon-lib

- https://www.phylopic.org/images

- https://journals.plos.org/plosbiology/article?id=10.1371/journal.pbio.3002395

6) We note that your Data Availability Statement is currently as follows: "Yes, this study contains human subject data. A statement in the Materials & Methods section confirms that informed consent was obtained and identifies the committee that approved the experiments.". Please confirm at this time whether or not your submission contains all raw data required to replicate the results of your study. Authors must share the “minimal data set” for their submission. PLOS defines the minimal data set to consist of the data required to replicate all study findings reported in the article, as well as related metadata and methods (https://journals.plos.org/plosone/s/data-availability#loc-minimal-data-set-definition).

7) Please amend your detailed Financial Disclosure statement. This is published with the article. It must therefore be completed in full sentences and contain the exact wording you wish to be published.

8) Your current Financial Disclosure states, "Yes ↳ Please add funding details. LOEWE Zentrum DRUID:Peter Kolb,Ger van Zandbergen D3 and Core facility; German Research Foundation SPP2225:Ute Distler,Stefan Tenzer,Ger van Zandbergen 446496937 and 446605368 ↳ Please select the country of your main research funder (please select carefully as in some cases this is used in fee calculation). GERMANY - DE".

However, your funding information on the submission form indicates different funders.

Please indicate by return email the full and correct funding information for your study and confirm the order in which funding contributions should appear. Please be sure to indicate whether the funders played any role in the study design, data collection and analysis, decision to publish, or preparation of the manuscript.

**Reviewers' Comments:**

Reviewer's Responses to Questions

**Part I - Summary**

Reviewer #1: This is a revised version of a manuscript that describes the role of p1/s1 nucleotidases/nucleases have in modulating the immune response to Leishmania. They use a cutting edge approach to delete a large region of the genome to delete the p1/s1 array and show that this array is important for digesting NETs. They additionally show that Leishmania utilizes adenosine to reduce the immune reponse.

I previously reveiwed this manuscript through Review Commons and they have responded well to my comments I had previously raised. Clarifications to the text and figures have been made and additional data has been included.

Reviewer #2: (No Response)

Reviewer #3: The authors responded to all my queries

**Part II – Major Issues: Key Experiments Required for Acceptance**

Reviewer #1: Generally, they have also responded appropriately to the comments from the other reviewers. However, I think there are some additional clarifications required:

- For the additional wording in the methodology (lines 99-101) in which they explain the composition of the stationary phase culture can a reference to figure S1 be included for the apoptotic-like cells and a representative set of PNA data shown for the metacyclic rate.

- For the comment about the compensatory mechanism, I looked at the data presented in the thesis the authors linked to and it did not seem to include RNAseq data from the KO cell lines. It just had expression levels for the different genes in the p1/s1 array in log, stationary phase and axenic amastigotes. The mechanism for compensation is not the focus of this manuscript so in my opinion it is not necessary to look at but I just wanted to clarify which data they were alluding to in that thesis.

- In response to Reviewer 3’s comment about the 50% of stationary phase that are dead or undergoing cell death, the authors state the in their culture conditions they see 50-60% dead parasites but in the new section of the methods they are calling this apoptotic-like. Similarly, in the response to Comment 5 that focuses on Figure 4F, the authors state that the stationary phase cultures have a comparable proportion of dead parasites (+/- 5%). In this context are the dead parasites the 50-60% apoptotic-like cells? Moreover, in the text (lines 371-381) the authors state that miltefosine induces 'early cell death processes' and stationary phase parasites 'undergo intrinsic cell death'. Can the authors clarify whether these are dead parasites or still alive but expressing apoptotic markers? There are a number of different phrases in the manuscript and the responses being used and I think it is important that there is consistency.

Finally, would the authors include an example gel to supplement Figure 5 which shows the endonuclease activity of the addback. They have it for the parentals and the KOs but not the addback. This could be included in a supplementary figure.

Reviewer #2: (No Response)

Reviewer #3: The authors responded to all my queries

**Part III – Minor Issues: Editorial and Data Presentation Modifications**

Reviewer #1: (No Response)

Reviewer #2: (No Response)

Reviewer #3: The authors responded to all my queries

PLOS authors have the option to publish the peer review history of their article (what does this mean?). If published, this will include your full peer review and any attached files.

**Do you want your identity to be public for this peer review?** For information about this choice, including consent withdrawal, please see our Privacy Policy.

Reviewer #1: No

Reviewer #2: No

Reviewer #3: No

**Figure resubmission:**
---

## [Editor Report · Decision Letter 1]

24 Apr 2026

Dear Dr. van Zandbergen,

We are pleased to inform you that your manuscript 'p1/s1, a 3’-nucleotidase/nuclease, allows Leishmania major to circumvent host innate immune response mechanisms' has been provisionally accepted for publication in PLOS Pathogens.

Best regards,

Dominique Soldati-Favre

Section Editor

PLOS Pathogens

Dominique Soldati-Favre

Section Editor

PLOS Pathogens

Sumita Bhaduri-McIntosh

Editor-in-Chief

PLOS Pathogens

orcid.org/0000-0003-2946-9497

Michael Malim

Editor-in-Chief

PLOS Pathogens

orcid.org/0000-0002-7699-2064
---

## [Editor Report · Acceptance letter]

Dear Dr. van Zandbergen,

We are delighted to inform you that your manuscript, "p1/s1, a 3’-nucleotidase/nuclease, allows Leishmania major to circumvent host innate immune response mechanisms," has been formally accepted for publication in PLOS Pathogens.

Best regards,

Sumita Bhaduri-McIntosh

Editor-in-Chief

PLOS Pathogens

orcid.org/0000-0003-2946-9497

Michael Malim

Editor-in-Chief

PLOS Pathogens

orcid.org/0000-0002-7699-2064